

# The 2022 Drought Needs to be a Turning Point for European Drought Risk Management

**Riccardo Biella** [1, 2], Anastasiya Shyrokaya [1, 2], Monica Ionita [3, 4], Raffaele Vignola [5, 6], Samuel Sutanto [7], Andrijana Todorovic [8], Claudia Teutschbein [1, 2], Daniela Cid [9, 10,] Maria Carmen Llasat [11, 12], Pedro Alencar [13], Alessia Matanó [14], Elena Ridolfi [15], Benedetta Moccia [15], Ilias Pechlivanidis [16], Anne van Loon [13], Doris Wendt [17], Elin Stenfors [1, 2], Fabio Russo [14], Jean-Philippe Vidal [18], Lucy Barker [19], Mariana Madruga de Brito [20], Marleen Lam [21], Monika Bláhová [22, 23], Patricia Trambauer [24], Raed Hamed [13], Scott J. McGrane [25, 26], Serena Ceola [27], Sigrid J. Bakke [28], Svitlana Krakovska [29, 30], Viorica Nagavciuc [3, 4], Faranak Tootoonchi [31], Giuliano Di Baldassarre [1, 2], Sandra Hauswirth [32], Shreedhar Maskey [33], Svitlana Zubkovych [2], Marthe Wens [13], Lena M Tallaksen [34]

Centre of Natural Hazards and Disaster Science, Uppsala, Sweden
Department of Earth Sciences, Uppsala University, Uppsala, Sweden
Paleoclimate Dynamics Group, Alfred Wegener Institute Helmholtz Center for Polar and Marine Research, 27570 Bremerhaven, Germany
Forest Biometrics Laboratory – Faculty of Forestry, "Ștefan cel Mare" University of Suceava, Universității street, no.13, 720229, Suceava, România
Water System and Global Change, Wageningen University and Research, Wageningen, the Netherlands
Gund Institute for the Environment, Vermont University, USA
Earth Systems and Global Change, Wageningen University and Research, Wageningen, the Netherlands
University of Belgrade, Faculty of Civil Engineering, Institute for Hydraulic and Environmental Engineering
Department of Civil and Environmental Engineering, Universitat Politècnica de Catalunya, Spain
Hydrogeology Group (UPC-CSIC), Spain
Department of Applied Physics, University of Barcelona, Spain
IdRA, Water Research Institut, University of Barcelona, Spain
Chair of Ecohydrology, Technical University of Berlin, Germany
Institute for Environmental Studies (IVM), Vrije Universiteit Amsterdam, The Netherlands
Dipartimento di Ingegneria Civile, Edile e Ambientale, Università degli Studi di Roma La Sapienza, 00184 Roma, Italy
Swedish Meteorological and Hydrological Institute, Norrköping, Sweden
Cabot Institute for the Environment, Bristol, UK
INRAE, RiverLy, Villeurbanne, France
UK Centre for Ecology & Hydrology, Wallingford, United Kingdom
Department of Urban and Environmental Sociology, Helmholtz Centre for Environmental Research, Leipzig, Germany
Water Resources Management (WRM), Wageningen University & Research (WUR), Wageningen, the Netherlands
Global Change Research Institute CAS, Brno, Czech Republic
Mendel University in Brno, Brno, Czech Republic
Deltares, The Netherlands
Department of Economics, Strathclyde Business School, University of Strathclyde, Glasgow
Applied Physics Department, Stanford University, CA, USA
Department of Civil, Chemical, Environmental and Materials Engineering, Alma Mater Studiorum Università di Bologna, Bologna, Italy
Norwegian water and energy directorate, Oslo, Norway
Ukrainian Hydrometeorological Institute, Kyiv, Ukraine



45    30 National Antarctic Scientific Center, Kyiv, Ukraine
31 Department of crop production ecology, Swedish university of agricultural sciences, Uppsala, Sweden
32 Department of Physical Geography, Faculty of Geosciences, Utrecht University, Utrecht, the Netherlands
33 IHE Delft Institute for Water Education, Delft, the Netherlands
Department of Geosciences, University of Oslo, Oslo, Norway

*Correspondence to*: Riccardo Biella (riccardo.biella@geo.uu.se)

**Abstract**

The 2022 European drought has underscored critical deficiencies in European water management. This paper explores these shortcomings and suggests a way forward for European drought risk management.

Data for this study was gathered through a continent-wide survey of water managers involved in this event. The survey
collected 481 responses from 30 European countries and is comprised of 19 questions concerning sectorial impact in the regions of the responders and drought risk management practices of their organizations. Information from the survey is enriched with climate-related information to offer a comprehensive overview of drought risk management in Europe. Our research focuses on four key aspects: the increasing risk of drought, its spatial and temporal impacts, current drought risk management approaches, and the evolution of drought risk management across the continent.

Our findings reveal a consensus on the growing risk of drought, which is confounded by the rising frequency and intensity of droughts. While the 2022 event affected most of the continent, our findings show significant regional disparities in drought risk management capacity among the various countries. Our analysis indicates that current drought risk management measures often rely on short-term operational concerns, particularly in agriculture-dominated economies, leading to potentially maladaptive practices. An overall positive trend in drought risk management, with organizations showing increased awareness
and preparedness, indicates how this crisis can be the ideal moment to mainstream European-wide drought risk management. Consequently, we advocate for a European Drought Directive, to harmonize and enforce drought risk management policies across the continent. This directive should promote a systemic, integrated, and long-term risk management perspective. The directive should also set clear guidelines for drought risk management at the national level and for cross-boundary drought collaboration.

This study and its companion paper "*The 2022 Drought Shows the Importance of Preparedness in European Drought Risk Management*" are the result of a study carried out by the Drought in the Anthropocene network.

## 1    Introduction

Just a few years after the exceptionally severe 2018-2019 drought (Moravec et al., 2021), large parts of Europe faced another record-breaking drought in 2022. Summer temperatures set new records across the continent (Copernicus Climate Change
Service, 2022), exceeding previous extremes observed during 2003, 2015-16, and 2018-19 droughts (Rakovec et al., 2022). Dry weather persisted through spring, initially affecting hydrological systems in the Eastern Alps, followed by extremely dry



conditions, soil moisture deficits, and streamflow drought in Central and Southern Europe (Montanari et al., 2023; Bonaldo et al., 2023). In several countries, this prolonged and widespread situation led to increased water withdrawals and eventually restrictions on water use due to persistent hot and dry conditions in May, June and July (Avanzi et al., 2024; Bonaldo et al.,
2023; Toreti et al., 2022). The Mediterranean was particularly affected by a dry winter and spring, exacerbating deficits in soil moisture and river flow (Toreti et al., 2022), with wide-ranging impacts on society and nature (Faranda et al., 2023).

This study analyses the 2022 European drought, linking its physical characteristics with sectoral impacts and current drought risk management practices across Europe. To do so, it employs a detailed survey of European water managers involved in the response to the event. The findings from the survey are used to discuss the limitations of the European drought risk management
framework and present recommendations for its improvement. This work is the result of a collaboration of the Drought in the Anthropocene (DitA) network (https://iahs.info/Initiatives/Scientific-Decades/helping-working-groups/drought-in-the-anthropocene/) and is a follow-up to the paper "*Lessons from the 2018–2019 European droughts: A collective need for unifying drought risk management*" (Blauhut et al., 2021). Following this study, a companion paper titled "*The 2022 European drought shows the importance of preparedness*" (Biella et al., 2024b) delves deeper into the need for preparedness measures in drought
risk management and is available in the same issue.

## 1.1 Drought in Europe

Droughts are periods of extraordinary water deficit in the hydrological cycle (IPCC, 2021; Van Loon et al., 2016), which impact can have adverse effects on the Socio-Ecological Systems (SES) (Van Loon et al., 2016). The possibilities of damage and losses to society and ecosystems caused by a given drought is referred to as drought risk (IPCCa, 2022; Hagenlocher et
al.,2023), which depend on the interactions between the severity of the drought event and the vulnerability and exposure of the SES (UNDRR, 2021). Finally, water scarcity indicates insufficient water availability compared to the demand of the SES (Van Loon et al., 2016), which can be the result of droughts, but can also be the result of human factors.

Droughts present complex challenges that not only affect the hydrological system, but rather affect all complex interdependencies of the SES (Kallis, 2008, van Loon et al., 2016). In Europe, this gives rise to a rippling effect of the drought
impact across terrestrial and aquatic ecosystems, economic sectors, and even the socio-cultural system (Crausbay et al., 2017), exacerbated by the increasing population and reliance on water in drought-affected areas. There is a close link between drought, heatwave, and wildfires, as evidence shows increasing heatwave and wildfire occurrences and severity with increasing dryness (Sutanto et al., 2020; Rodrigues et al., 2023). This compounding hazard presents new threats to human and the environment.

Climate change is intensifying drought hazard over most of the continent (Spinoni et al., 2018; Jaagus et al., 2021). While
recurrent droughts have historically been a major concern for water-scarce regions such as the Mediterranean basin, recent decades have witnessed a consistent increase in their frequency, especially during summer (Caloiero et al., 2018; Markonis et al., 2021; Montanari et al., 2023). Combined with an ever-increasing demand for water globally (Savelli et al., 2023), water scarcity has become an emerging threat to European countries that requires revisiting water management strategies (Stein et al., 2016). Continental and northern regions of Europe are not spared from drought, as studies show increasing drought risk



even in Western Europe and Northern Scandinavia (Spinoni et al., 2018) as well as in Eastern Europe particularly Ukraine (Semenova & Vicente-Serrano, 2024). Yet, the Mediterranean remains the most drought-prone region in Europe (Caloiero et al., 2018, 2021) and a global hot spot now and in the future (IPCC, 2022a). In their study, Ionita & Nagavciuc (2021) underscore the significant role of rising temperatures in the increasing frequency of droughts due to increased evapotranspiration and reduced snow accumulation. The record temperatures and altered rainfall patterns that are already being observed confirm the

increasing drought risk in Europe due to global warming (Stagge et al., 2017; Vicente-Serrano et al., 2010), which is likely to be aggravated even further in the future (Faranda et al., 2023; Schumacher et al., 2022, 2024), leading to drier and more intense droughts (Ionita et al., 2022). In fact, more than 30% of all extreme droughts observed in Europe since 1950 have occurred in between 2012 and 2022 (van Daalen et al., 2022). This staggering sequence of unprecedented extreme droughts stresses the importance of comprehensive and wider drought governance frameworks to enable effective drought risk management,

including mitigation, adaptation, preparedness, and early warnings (Blauhut et al., 2021).

**1.2 Systemic drought risk management**

Drought in Europe often cover large regions and last for few to several months, causing severe socio-economic impacts on different areas of the SES, including agriculture, water supply, water quality, energy production, ecosystems, public health, tourism, and recreation. Such impacts evolve slowly and tend to have co-occurring (Shyrokaya et al., 2023) and cascading

effects across sectors in different parts of the SES (de Brito, 2021). In recent decades, annual drought-related economic losses in the EU and UK have been estimated at around €9 billion, with the highest losses in Spain (€1.5 billion/year), Italy (€1.4 billion/year), and France (€1.2 billion/year) (Cammalleri et al., 2020). Climate change is projected to further increase these losses, with a projected €65 billion for the EU and UK combined by 2100 (Naumann et al., 2021). Depending on the region, 39-60% of these losses are in the agriculture sector, while 22-48% are in the energy sector (Cammalleri et al., 2020). Atlantic

and Mediterranean Europe experience the highest drought-related economic losses, contributing to 68% of total European losses in recent years, a share that is expected to rise with increasing temperatures, potentially reaching 85% with 3°C of warming (Cammalleri et al., 2020). The European Drought Risk Atlas (Rossi el al., 2023) demonstrated that extreme impacts on ecosystems and inland navigation are more often reported than those on other sectors. They estimate the effect of droughts on aquatic and terrestrial ecosystems is the highest in Finland and Croatia respectively, and note the largest increases in drought

risk for the ecosystem in Italy and Spain. Yet, these figures represent only the quantifiable part; indirect impacts are often non-monetary and challenging or even impossible to quantify. Consequently, the true extent of drought impacts is likely much larger than current estimates suggest.

Droughts, unlike many other disasters, have diffuse beginnings and endings (van Loon et al., 2024), involving various components of the SES as they propagate through the hydrological cycle affecting a wide range of sectors. Traditional drought

management approaches have largely been reactive, focusing on crisis management rather than risk management, which often results in ineffective and poorly coordinated risk management. However, researchers have warned against short-sighted measures, as this can often lead to unintended consequences and maladaptation (Biella et al., 2024; Di Baldassarre et al., 2017;



Magnan et al., 2016), increasing the vulnerability of the system to future droughts. Instead, researchers argue that droughts necessitate a systemic perspective that recognizes the complexities and interlinkages between elements and processes of the
systems (Hagenlocher et al., 2023; Kallis, 2008; Van Loon et al; Wilhite, 2019), often underscoring the risks associated with infrastructural measures (Di Baldassarre et al., 2017), and the benefits provided by ecosystem-based adaptation (IPCC, 2022a; McVittie et al., 2018; Sudmeier-Rieux et al., 2021; Vignola et al., 2009). This perspective calls for continuous monitoring and adaptive management strategies that can respond dynamically to changing conditions and focus on dependencies, non-linearities, feedback dynamics, compounding and cascading effects, tipping points, multi-level risk, and deep uncertainties
(Hagenlocher et al., 2023, de Brito et al. 2024).

One such example is Integrated Drought Management (IDM), which advocates for a proactive, risk-based approach that incorporates monitoring, early warning systems, and vulnerability assessments (Grobicki et al., 2015; Wilhite, 2019). IDM strategies are essential for mitigating drought impacts, balancing water demand and supply while ensuring environmental sustainability (Wendt et al., 2021). Regional governments and local communities play an essential role in drought management,
as drought resilience depends on the collective capacity of stakeholders across different scales (Kchouk et al., 2023). This underscores the need for policies that are sensitive to local contexts and that can mobilize resources and knowledge effectively across various governance levels. The Integrated Drought Management Programme (IDMP) launched by the Global Water Partnership (GWP) emphasizes regional cooperation and capacity building. The IDMP aims to create a coordinated framework for drought monitoring and management that involves several decision-making levels, from government officials to local
stakeholders (Bokal et al., 2014; WMO & GWP, 2014). Finally, integrated and systemic drought risk management needs to account for the interplay between other forms of disaster risk management and drought risk management. In particular, research has demonstrated the interaction between flood and drought risk management, and the need for holistic approaches that manage the two hazards ((Barendrecht et al., 2024; Di Baldassarre et al., 2017). Systemic approaches to drought risk management are crucial for developing effective drought policies and plans that can adapt to the region's specific needs.

**1.3 European drought governance framework**

Overall, European drought awareness and preparedness have been following an upward trajectory, as drought governance has been mainstreamed across many countries (Biella et al., 2024b; Kreibich et al., 2022; Publications Office of the European Union, 2023).
However, this progress has not been a steady one. Awareness of drought risk peaks shortly after an extreme event, capturing the attention of the public and policy makers, potentially leading to changes in water governance and drought risk management.

*"Water is not a commercial product like any other, but rather, a heritage which must be protected, defended and treated as such"*

European Commission
(DIRECTIVE 2000/60/EC Preface, Comma 1)

The scale of Europe's droughts and its interconnected socio-hydrological systems necessitate continent-wide drought risk
management. Yet, Europe (where the EU represents the largest governance body) lacks a unified drought policy, relying



instead on other water-related directives and non-binding communications (Hervás-Gámez & Delgado-Ramos, 2019; Publications Office of the European Union, 2023; Stein et al., 2016). The 2000 European Commission's (EC) *Water Framework Directive* (WFD) is considered one of the most ambitious and substantial pieces of legislation dealing with water resource management (Voulvoulis et al., 2017), instituting catchment-level water management, environmental output requirements, unified monitoring, and international collaboration for transboundary catchments (Publications Office of the European Union, 2023; Stein et al., 2016). The WFD promotes a precautionary approach, emphasizing water conservation and stating that water is "a heritage to be protected" (DIRECTIVE 2000/60/EC Preface, Comma 1). Further developments on drought risk management include the 2007 EC *Communication on Water Scarcity and Droughts* and the 2012 EC communication *Blueprint to Safeguard Europe's Water Resources*, which provide guidelines for Drought Management Plans (DMPs) and country-level drought risk management (Hervás-Gámez & Delgado-Ramos, 2019). The latter is particularly important for its emphasis on water conservation, stating the need to prioritize demand-reduction over efficiency measures, and especially over increased supply and/or infrastructural measures (Hervás-Gámez & Delgado-Ramos, 2019; Stein et al., 2016). Despite its ambitions, the EU's drought risk governance framework has significant gaps: the WFD does not directly address drought risk management, and the EC Communications of 2007 and 2012 are non-binding, lacking mandatory action for member states (Publications Office of the European Union, 2023; Stein et al., 2016). In contrast, the EC established a *Flood Directive* in 2008, which laid the basis for European-level flood governance, creating a precedent for integrated Europe-wide guidance on a hazard-specific risk management policy.

### 1.4 Objective of this research

The current drought governance framework offered by the EU is not suited for managing the increasing drought risk that Europe is experiencing (Publications Office of the European Union, 2023; Stein et al., 2016). In this study, we provide an overview of the 2022 European drought, demonstrating the linkage between its physical aspects, sectoral impacts, as well as adopted risk management measures. In this paper, we explore four main questions: (1) "*Is drought risk increasing?*"; (2) "*What is the spatial and temporal evolution of drought impacts in 2022?*"; (3) "*What are the drought risk management measures in place in 2022?*"; and (4) "*How is drought risk management changing across Europe?*". To answer these, we employ a large survey targeting water managers across Europe (described in Sec. 2.2). The survey results show the ramifications of drought impacts across Europe and provide insights into the status and trends in drought risk management (Sec.3.1, Sec 3.2, and Sec. 3.3). Additionally, two case studies (referred to as "regional spotlights") are used to provide additional insights on various aspects of drought risk management during the 2022 event, combining results from the questionnaire with additional information (Sec. 3.4). In the discussion (Sec. 4) we underline the need for unified drought risk management coordination at the continent level. Following up on the plea made by Blauhut et al. (2021), we advocate for the development of an EC *Drought Directive*, inspired by the EC *Flood Directive* of 2007 (Sec. 5). This directive should offer a legally-binding policy mix that enshrines into law integrated and systemic drought risk management, placing equity, sustainability, and environmental needs at its centre.



## 2   Methods and data

### 2.1 Climate data

Meteorological drought refers to a prolonged period of abnormally low precipitation, leading to water deficits. The Standardized Precipitation-Evapotranspiration Index (SPEI; Vicente-Serrano et. al., 2010) measures meteorological drought by considering both precipitation and potential evapotranspiration. Similar to the Standardised Precipitation Index (SPI; McKee et al., 1993), it relies on selecting a probability distribution to normalise the index, allowing for comparisons across climates (Stagge et al., 2015). Positive SPEI values indicate wetter conditions, while negative values suggest meteorological drought. Shorter accumulation periods of SPEI (e.g., SPEI-1 and SPEI-3) are used as proxies for meteorological and agricultural droughts, while longer accumulation periods (e.g. SPEI-6 and SPEI-12) are taken to represent hydrological drought. Seasonality of drought is indicated using SPEI-3 winter is defined by calendar  months from December to February, with the SPEI-3 for Februarys used to assess this period (representing the wet/dry anomaly three months back, i.e., the calendar winter months). Similarly, SPEI-3 in May represents the spring (March to May), SPEI-3 in August represents the summer (June to August), and SPEI-3 in November represents the autumn (September to November).

SPEI estimates derive from monthly precipitation (PP), mean air temperature (TT), and potential evapotranspiration (PET) based on the data from the Climatic Research Unit (CRU) TS v. 4.07 dataset (Harris et al., 2020), with a spatial resolution of $0.5° \times 0.5°$., The climatological period 1971 – 2000, is used here as a reference period for the computation of SPEI. To this end, we use the R package "SPEI" (https://cran.r-project.org/web/packages/SPEI/index.html), which is based on the probability distribution of the difference between PP and PET. Data are normalized into a log-logistic probability distribution to derive the SPEI values (Vicente-Serrano et al., 2010). Potential evapotranspiration is determined using the Penman-Monteith equation (Vanderlinden et al., 2008).

### 2.2 Impact and management data

### 2.2.1 Questionnaire targeting water managers

To collect data on drought impact and management measures, we designed a questionnaire targeting water managers responding to the 2022 European drought. Designed by a team of researchers with expertise in drought risk management belonging to the DitA working group of the HELPING initiative the questionnaire (hereafter referred to just as *the questionnaire*) covers a wide range of topics. These include sectoral impacts of the drought, the occurrence of compounding and concurrent hazards, the measures taken by the respondents' organizations (along with their effectiveness and timeliness), the presence and use of preparedness measures, and developments in drought risk management across Europe. The survey is comprised of 19 questions (24 including additional clarification options) out of which 14 (17 including clarifications) are analysed in this study. The selected questions (listed in Table 1) focus specifically on drought impacts and drought risk management, as well as general questions regarding the respondents' organizations background and function.



To minimize misunderstandings, a glossary of terms like "drought risk management," "drought risk," and "drought risk management plans" is provided at the beginning of each relevant questionnaire section. Translated into 19 languages (listed in supplement, Sec. S1.2), the questionnaire was distributed from March to October 2023. The sampling strategy utilized the DitA group's network. Key contacts in various European countries received the questionnaire through personal connections or web searches for experts, academics, and public organization contacts, who then disseminated it further via snowball sampling

(or chain sampling). No personal information was automatically collected, ensuring compliance with the General Data Protection Regulation (GDPR). An overview of the survey is provided in the supplement (Sec. S1).

**Table 1:** Questions included in the questionnaire to water-managers used in this study. *Question number* refers to the number of the question in the questionnaire. Type of question indicates the typology of the question. *Open-ended questions* had a field where the respondents could leave their answers in text. *Multiple choices* were provided as either single, or matrixes (i.e. multiple choice for multiple options of the

question). *Multiple choice + open-ended* means that the multiple choice have one option that can be answered as an open-ended question if selected.

| Question number | Question | Type |
|---|---|---|
| 1 | What type of organization do you belong to? | Multiple choice |
| 2 | At which level does your organization operate? | Multiple choice |
| 3 | In which country is your organization located? | Multiple choice + open |
| 4 | In which municipality/region do you operate (name, region, country)? | Open-ended question |
| 6 | Which sectors does your organization operate in? | Multiple choice + open |
| 7 | How severe was the impact of the 2022 drought on a scale from 1 (Not affected) to 5 (Severe)?  [by sector] | Multiple choice matrix |
| 9a | When were the impacts first seen (month)?  [by sector] | Multiple choice matrix |
| 9b | When did the 2022 drought end (month)? [by sector] | Multiple choice matrix |
| 10 | Which sectors were prioritized in the distribution of water resources? [by sector] | Multiple choice |
| 12 | What were the main measures taken by your organization? | Open-ended question |
| 13 | When did your organization first take measures to mitigate the impact of the 2022 drought? | Multiple choice |
| 14 | How effective were the measures taken? | Multiple choice |
| 17 | Compared to the 2018-2019 drought, your organization was…   [More; Less; Same]; [Aware; Prepared; Effective in response] | Multiple choice matrix |
| 18a | Do you think that the risk posed by droughts is… [Increasing; Same; Decreasing] | Multiple choice matrix |
| 18b | Elaborate (optional) | Open-ended question |
| 19a | Do you expect the drought to become a more significant risk to manage for your organization in the future? | Multiple choice |
| 19b | If yes, how is drought management changing in your organization (optional) | Open-ended question |



**2.2.2 Dataset**

The survey gathered 487 responses, of which 481 are deemed valid. Invalid responses include those from outside the study area, or clear duplicates. Responses were received from 30 European countries, predominantly from Italy, Sweden, Croatia,

Romania, and Serbia. Fifteen countries each have ten or more respondents, collectively accounting for 89% of the total responses (Fig. 1b). Notably high concentrations of responses, came from the Alpine and Adriatic regions, central Balkan Peninsula, Rhine Valley, Southern Sweden, and the Pyrenees Region. A significant portion of respondents (76%) are employed in public and governmental organizations, while the remainder work in private companies (8%), research institutes (7%), NGOs (4%), and unspecified organizations (4%). Regarding their operational scope, a majority (65%) operates regionally,

with 27% at the national level and 6% internationally (Fig. 1a). Respondents are involved in various sectors critical to drought management, especially water quality, public water supply, and agriculture, with 226, 220, and 189 responses, respectively, in addition to several ecosystem-related areas (Fig. 1c). Sectors are based on the classes of drought impact as defined in the European Drought Impact Inventory (EDII, Stahl et al., 2016). The same classes were used to indicate drought impact by sector.

As a survey-based dataset, several limitations are inherent. First, responses reflect respondents' subjective views. Metrics describing drought risk management, such as response effectiveness or awareness, should thus be understood as perceived effectiveness and awareness. However, given the respondents' professional role in drought risk management, their subjective observations are considered relevant and valid metrics. Second, snowball sampling, while necessary here, limits the representativeness of the data, particularly when aggregated into sub-groups (such as by country, area of operation, or type of

organization). This study assumes that the sub-group samples are representative of their larger groups, though this cannot be confirmed. Finally, since different countries received vastly different numbers of responses, comparison between countries is limited. To reduce bias, the study only displays information from countries that received more than 10 valid responses to a specific question. Additionally, information is also presented at the regional level. The regions are adapted from those used in *The World Factbook* (https://www.cia.gov/the-world-factbook/), then adjusted with Southern Europe divided into two parts

instead of three to keep a more equal distribution of the responses: consequently, Italy, Vatican, and San Marino are grouped with the South Western region, and Greece and Cyprus with the South Eastern region. In the figures, the regions will be referred to using acronyms: norther-western (NW), north-eastern (NE), western (W), central (C), eastern (E), south-western (SW) and south-eastern (SE). An overview of the regions used in this study and the acronyms used for each country can be found in Sec. S2 of the supplement.



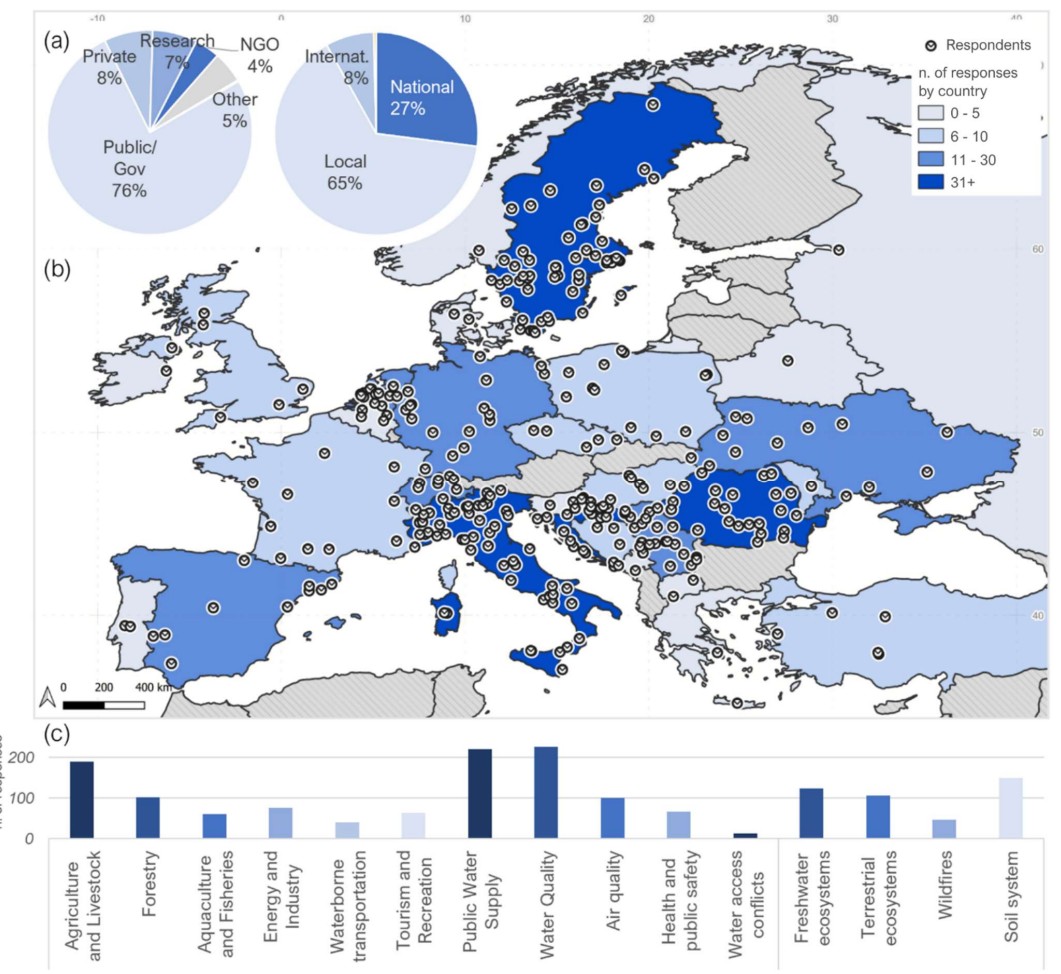

**Figure 1:** Overview of the distribution of respondents by (a) type of organization and operational level, (b) location and frequency across Europe, and (c) areas of operations. The total number of responses to the questionnaire was 481.

### 2.2.3 Drought impact

Questions 9a and 9b required respondents to indicate the beginning and end of the observed drought impact for each sector.
Responders could leave blanks for sectors they were not familiar with, and only provide indications for the sectors in which they were knowledgeable. The start and end of the observed impact are reported by month. The observation period extends for nine months; from March to September 2022. Additionally, the option "before March 2022", and "after September 2022" are



available. Respondents were required to indicate the severity of the impact by sector on a scale from 1 (not severe) to 5 (very severe) (question 7, see Table 1), as well as the prioritization that each sector received in the response to the drought as "low

priority", "medium priority", or "high priority" (question 9, see Table 1). Again, respondents could leave blank any sector for which they could not provide a response.

### 2.2.4 Drought risk management

Respondents detailed the drought risk management measures taken by their organizations in an open-ended question (question 12, Table 1). The responses to this question were reclassified using a typology devised by Reckien et al. (2023), which in turn

is based on the IPCC AR6 GAMI (Ch. 16) (IPCC, 2022b). To facilitate reclassification, responses were first translated from their original language to English using ChatGPT (chatgpt.com). These translations were then validated by native speakers. Additionally, the responses to the question on measures taken were also classified based on the recommendations outlined in the EC *Blueprint to Safeguard Europe's Water Resources* of 2012. The Blueprint prioritizes the different types of drought risk management measures, placing demand decrease as the highest priority, followed by prioritization and efficiency measures,

and assigning the lowest priority to supply increase measures and infrastructural measures. Based on this, three categories were established to evaluate responses in this study: "Decrease Demand"; "Prioritization and Efficiency"; "Increase Supply". Demand-side measures refer to measures aimed at reducing water demand in order to match the decreased supply. Conversely, supply-side measures attempt to integrate water supply through additional or alternative sources of water (e.g. groundwater) to meet demand. To ensure the validity of the classification, two researchers were involved in the classification process, and

an agreement test was carried out, which reached 86% and was considered satisfactory. Details regarding the agreement test are reported in the supplement (Table S2).

Question 14 prompted respondents to rate the effectiveness of the measures taken during the 2022 drought on a scale from 1 (not effective) to 5 (very effective). Respondents also had the option to leave the question blank (reported as "no answers" or "NA"), or indicate "I don't know". Additionally, respondents could respond "not relevant". This option was originally intended

for respondents whose actions, like monitoring and data collection, do not directly impact drought management, or for those who took no measures. To prevent misinterpretation (inferring "not relevant" as "not effective at all"), this option (less than 1) was placed separately from the 1-to-5 scale in the questionnaire.

Finally, question 17 allowed respondents to indicate the direction of drought risk management for their organizations by indicating whether their organizations were more, less, or equally aware of drought risk in 2022 compared to 2018 (i.e. the

year of the previous large-scale European drought). They were also asked to assess the preparedness and effectiveness of their organization's management in 2022 compared to 2018.



## 3 Results

### 3.1 Drought occurrence

### 3.1.1 Development of the 2022 drought in Europe

The onset of the 2022 drought was already visible in the winter of 2021-2022 (Fig. 2a), which was unusually warm and dry across the southern and eastern parts of Europe. The Alps, a crucial source of freshwater for the continent, received significantly less snowfall than average (Carrer et al., 2023; Montanari et al., 2023). Snowpack and seasonal snow cover act as natural reservoirs, slowly releasing water throughout the spring and summer. The absence of snow means that rivers and streams are deprived of their usual replenishment, leaving them vulnerable as temperatures and evaporative demand rose,

especially for the Rhine and Danube rivers, which are essential rivers for inland waterways navigation and drinking water (Van Loon et al., 2024).

Spring 2022 was characterized by warm conditions across most of Europe (Faranda et al., 2023), further exacerbating the dry conditions (Fig. 2b). The most affected countries were France, Italy, Germany, Poland, Czech Republic and the Balkan countries. High temperatures accelerated evaporation from soils and water bodies, impacting ecosystems, and increasing the

demand for irrigation. This early onset of warm and dry weather imposed stress on already depleted water resources, raising concerns for the months ahead. In summer, a high-pressure system persisted over Europe, creating a heat dome that trapped warm air and blocked moisture-bearing weather systems (Bakke et al., 2023). Temperatures soared to record highs, drying out soils, wilting crops, and fuelling wildfires. The summer period witnessed the drought moving northward affecting the UK and the Republic of Ireland, the Netherlands, Germany, Northern Poland, Belarus, and Ukraine (Fig. 2c), whereas the most northern

parts remained unaffected, except for southern Scandinavia in the spring. In autumn (Fig. 2d) only a small region was still affected by extremely dry conditions, namely the southern and eastern parts of Spain, Turkey, Greece and Bulgaria, the north-eastern part of Germany and the western part of Poland, and the Baltic states.

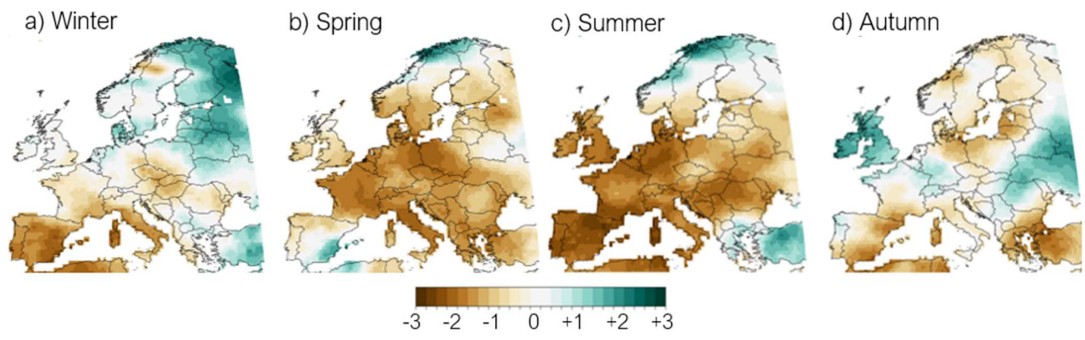

**Figure 2:** Seasonal evolution of the Standardized Potential Evapotranspiration Index for 3-month accumulation (SPEI-3) for 2022; a) Winter
(SPEI-3 February); b) Spring (SPEI-3 May); c) Summer (SPEI-3 August) and d) Autumn (SPEI-3 November).




The SPEI-6 for September, which indicates the wet/dry condition over the growing season, for four major droughts in Europe (i.e., 2003, 2015, 2018, and 2022) of the last two decades is shown in Fig. 3 (top row). The top-seven ranking of the lowest SPEI-6 for these events is shown in Fig.3 (bottom row). A rank of one means that SPEI for a given year is record-breaking, i.e., the lowest during the analysed period. The location of the drought, the size of the affected area, and its unprecedented
level, vary across the events. For example, in 2003, the core of the drought, in terms of record-breaking values, was in central Europe (Fig. 3a and 3b), whereas in 2015, the most affected regions were in the eastern part of Europe (e.g., eastern Poland and Ukraine) (Fig. 3c and 3d). In 2018, the core of the drought was over Germany, Poland and the southern part of Sweden (Fig. 3e and 3f), while between March and September 2022, northern Spain and south-western France were affected by record-breaking meteorological droughts. None of the events covered the whole of Europe, but in most cases, more than 50% of the
continent experienced at least moderate drought conditions. It is worth noting that the two most recent events (2018 and 2022) show the highest continuous area of record high SPEI values (dark brown colour).

From a hydrological perspective, in 2022, many prominent European rivers, including the Rhine, Danube, Po, and Ebro experienced one of the most severe droughts in recent decades (more details in Fig. S3 in the supplement). By the end of winter 2022, these rivers faced prolonged drought conditions that persisted until the end of summer for the Rhine and Danube, and
even longer for the Po and Ebro.

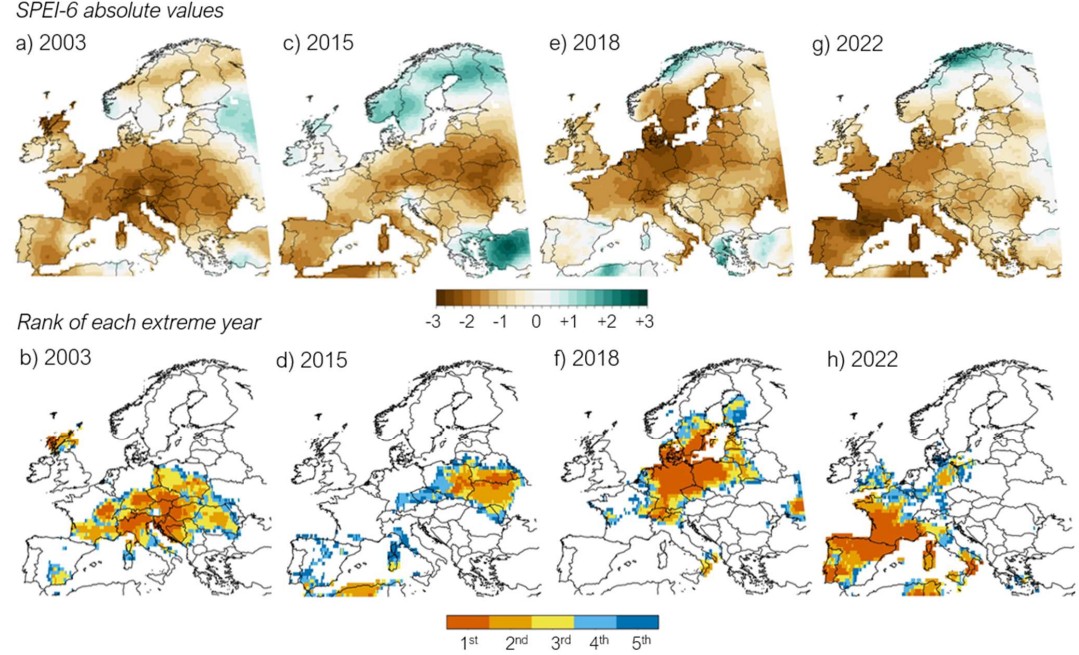




**Figure 3:** Comparison of the SPEI6 for September of the main Europe-wide droughts since 2000: Namely, 2003, 2015, 2018, and 2022 (top row) and their associated ranking (bottom row). The period analysed is 1950–2022. A rank of one signifies that SPEI6 September for a given year (i.e. 2003, 2015, 2018 or 2022) is record-breaking, i.e. the lowest value during the analysed period.

Over the last 70 years, droughts have become more frequent, more extreme, and more extensive over Europe (Fig. 4). Moderate (SPEI between -1 and -1.5) and severe (SPEI between -1.5 and -2) droughts, in particular, have intensified at the European level, especially after 2000's, in agreement with previous studies (Ionita and Nagavciuc, 2021). SPEI values lower than -2 are referred to as extreme drought (Vicente-Serrano et. al., 2010). As for the 2022 drought, this event was the most extreme that the continent has experienced since 1954. However, unlike the 1954 drought event, which mostly took place during winter,

recent droughts, and in particular the 2022 drought, occurred between April and October, and peaked in July and August (Fig. 4).

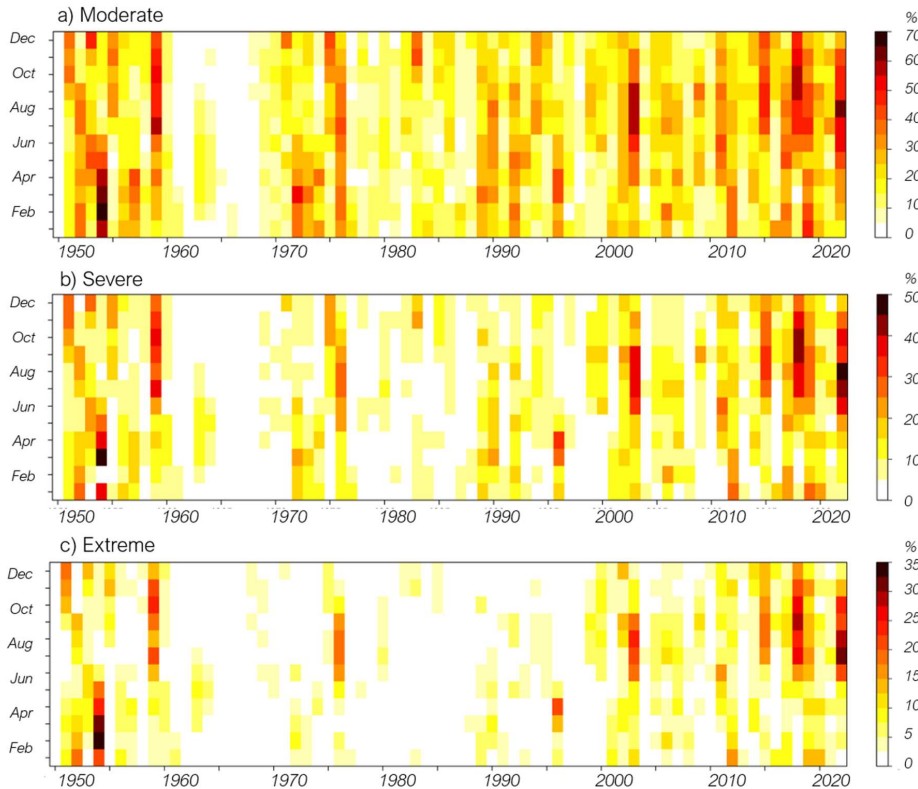

**Figure 4:** Temporal evolution of the percentage of area affected by droughts at European level for three drought severity categories: moderate (a; SPEI between -1 and -1,5), severe (b; SPEI between -1,5 and -2), and extreme (c; SPEI below -2). The colour of the cell indicates the percentage of the European area covered by a drought (i.e. below SPEI) of the corresponding intensity.



### 3.1.2 Perception of drought risk among the respondents

According to the survey, the vast majority of the respondents (87%) consider that the risk of drought has been increasing. In contrast, only 89% of the respondents consider that the risk remains the same, while just 1% of the respondents think that the risk is decreasing. Additionally, 2% of the respondents are unsure about the risk (i.e., responded "*I don't know*"). The highest

levels of concern for increasing risk are expressed by respondents operating in water management in the ecosystem-related fields (i.e. Terrestrial and Aquatic Ecosystems, Wildfires, and Soil System); 92% of those operating in Terrestrial Ecosystems, and 91% for those operating in Freshwater Ecosystems. Regarding economic sectors, the respondents within Energy and Industry considered drought risk to be increasing the most (92%), followed by those working on Air Quality (89%). On the other hand, the sectors where the least percentage of respondents think that the risk of drought is increasing are Tourism and

Recreation (83%), Human Health (84%), and Forestry (86%). Tourism and Recreation, Human Health, and Waterborne Transportation show the highest number of respondents who consider drought risk to be unchanged, with 14%, 13%, and 13%, respectively. At the country level (considering only countries with over 10 responses), countries with the highest percentage of respondents indicating that drought risk is increasing are France and the UK (both 100% out of 15 and 14 answers, respectively), Serbia (93% out of 29 answers) and the Netherlands (93% out of 28 answers). On the other hand, respondents

from Sweden (73%), Romania (74%), and Germany (79%) indicated the smallest increase in drought risk. Furthermore, countries with the largest share of respondents perceiving that the risk posed by drought is unchanged, are Sweden (21%), Romania (14%), Germany, and Switzerland (both 12%). The largest share of respondents indicating decreasing risk are from Sweden (5%), Croatia (2%), and Romania (2%).

### 3.2 Drought impact

### 3.2.1 Impact duration

According to the respondents, the impacts of the 2022 drought were initially observed in Southern Europe (Fig. 5a). In particular, most sectors in Spain showed signs of drought impacts before March 2022, while in Italy, Agriculture was the key sector displaying early signs of impact (before March). Central and Eastern Europe, such as Hungary, Poland and Ukraine, also exhibited an early onset of drought impacts, particularly in sectors like Agriculture, Fishery and, in some cases, Forestry.

The remaining part of Europe (excluding northern Europe not affected by the event) experienced the drought impact later, typically not earlier than June or July. However, some exceptions can be seen in certain sectors, notably Forestry, Energy and Industry, Tourism and Recreation, Air Quality, and Water Conflict, as well as in some countries where first drought impacts manifested later, i.e., after September. This aligns with expectations, as impacts on Forestry may require a longer period to become apparent, given that dieback results from prolonged dry conditions, diminishing pest and disease resistance over time

(Shyrokaya et al. 2023; Bastos et al. 2020; Messori et al. 2021; Wu et al. 2022). Additionally, hydropower production is dependent on reservoir storage, short and long-term weather forecasts, and the energy market, all of which are also influenced by droughts (Okkan et al., 2023). Survey reports also indicated that the drought ended last in Southern, Central, and Western



Europe (the Netherlands, Germany, and France). Consequently, Southern Europe emerges as the region experiencing the longest-lasting impacts, persisting for over nine months in some cases (the entire observation period covered by our questionnaire). This severity of drought impacts, as reported across much of Southern Europe, is mirroring the drought extent and severity as depicted by SPEI-3 and SPEI-6 indices (Fig. 5b, lower left panel), starting as early as March. Overall, many countries show lags between the drought hazard (represented by the SPEI) and impact occurrences, ranging from 0 months in Spain to 5 months in France. As for the drought termination, several countries reported drought impacts beyond the drought period as defined by SPEI-3 (Fig. 5b, lower right panel). The main reason being that rainfall occurring during summer may terminate meteorological and agricultural droughts, represented by SPEI-3, while hydrological droughts may persist longer depending on the memory of the hydrological system.

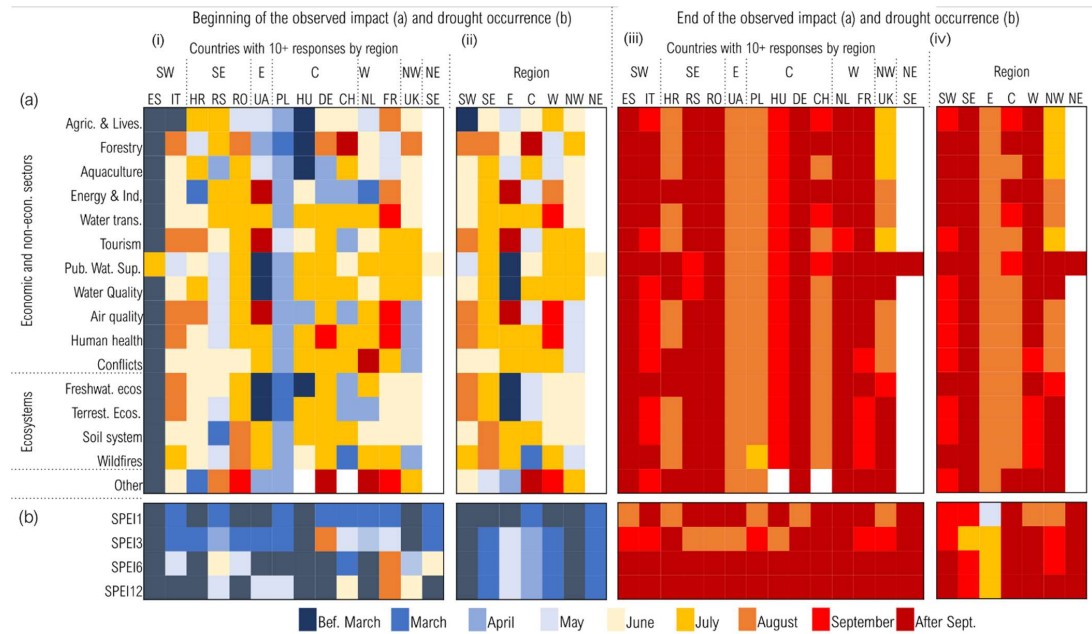

**Figure 5:** Most commonly reported beginning (left) and end month (right) of drought impacts in Europe listed from south to north (a). The lower plot (b) indicates the onset of the drought defined as the first month when more than 50% of the territory was under drought conditions (SPEI < -1) for SPEI-1, SPEI-3, SPEI-6, and SPEI-12 of each month. Only countries with 10 or more responses are shown. The European regions are described using the acronyms: SE (southeast), SW (southwest), E (east), C (central, W (west), N (Northwest), NE (northeast). Countries are indicated using their two-letter country code.

### 3.2.2 Impact severity and prioritization

In terms of impact severity, Southern Europe experienced earlier and longer-duration droughts with more severe consequences, compared to less severe impacts in the north. In particular, sectors such as Agriculture, Forestry, and Public Water Supply





were highly impacted in Central and Southern Europe, with increased wildfires and soil degradation as examples. Conversely, Northern Europe witnessed less severe impacts (Fig. 6), as expected due to the less severe drought conditions or even no drought at all.

Notably, certain highly-impacted sectors including Forestry in Germany as well as Energy and Industry and Fisheries in Italy, received low priority water allocation. In contrast, sectors with perceived milder impacts, such as Public Water Supply and Water Quality, Water Transportation, and Tourism and Recreation, were considered high priority by the respondents across multiple countries. Yet, less severe impacts can also result from the priority given to mitigation and adaptation measures. As prioritization affects drought impact, a high priority is likely linked to a less severe impact within certain sectors (e.g. Public Water Supply), and as such can be indicative of an effective response. Moreover, sectors where impacts generally take longer

to materialise, such as Forestry, are likely to receive lower prioritization (Fig. 6). Still, this prioritization discrepancy highlights the nuanced approach to managing diverse impact levels across sectors in response to varying degrees of drought and impact severity.

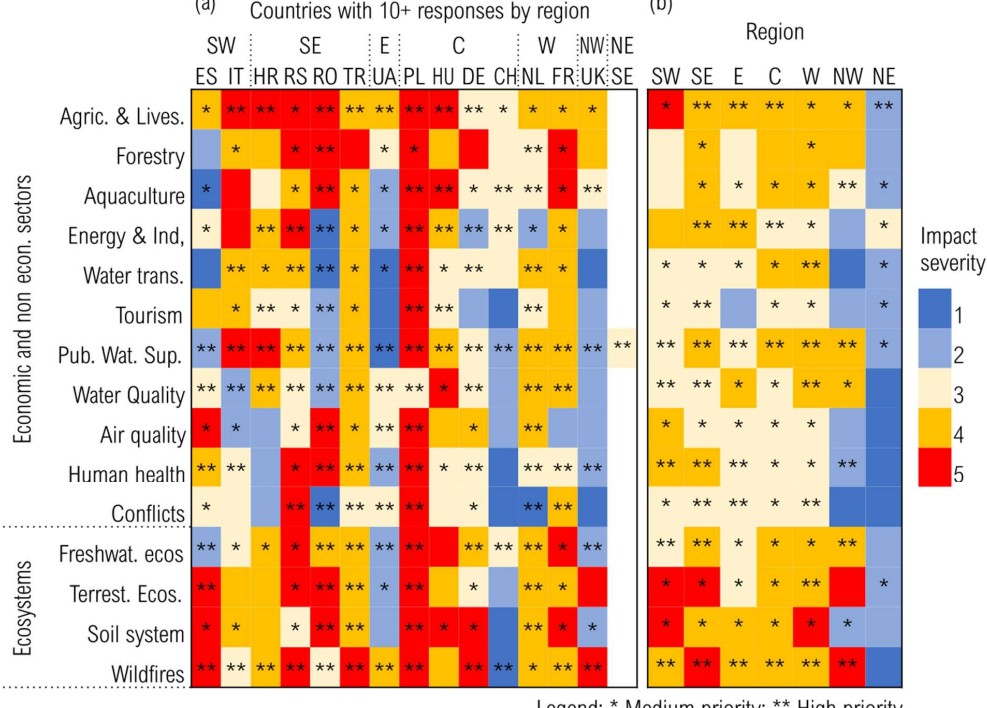

**Figure 6:** Impact severity (on a scale from 1 to 5; where 5 is the most severe level) on the various sectors and their prioritization according
to the respondents (on a scale from none, to 1 or 2 stars that indicate low-, medium- and high priority, respectively). Only countries with 10



or more responses are presented. Countries are grouped into geographical regions. The panel to the right shows the value for the entire geographical region. These include countries with less than 10 responses. The European regions are described using the acronyms: SE (southeast), SW (southwest), E (east), C (central, W (west), N (Northwest), NE (northeast). Countries are indicated using their two-letter country code.

**3.3 Drought risk management**

**3.3.1 Types of drought measures taken**

Most measures taken fall into two primary categories: those related to water supply (2%) (i.e., increasing sources to meet demand or prioritizing users, such as increasing the use of groundwater), and those concerning water use and demand (19%) (i.e., reducing demand to meet availability, such as by introducing restrictions on use) (Fig. 7). This trend is observed across
all sectors, although variations exist between countries. Countries in Southern Europe (and the Netherlands) tended to favour water supply management, while countries in Central and Western Europe predominantly focussed on water demand management. Other prevalent measures included awareness raising (19%), which was common across all sectors and many countries, and monitoring (9%), most common in the Public Water Supply and Water Quality sectors. Monitoring was also particularly notable in Sweden (where the drought was less severe). Germany, France, the Netherlands, and Croatia also
implemented 'incentive and compensations' schemes to tackle drought impacts. Farm-related management practices were common in Romania and Turkey. Ecosystem-based measures were only common in Poland, where many responses came from natural park management authorities, and were mostly missing in other countries (with some minor exceptions in the Balkan region). Sweden and Ukraine most frequently reported that few or no measures were taken. In Sweden, the milder manifestation of the drought led many respondents to deem drought management unnecessary (in agreement with the two respondents from
Norway). This is reflected in the Public Water Supply and Water Quality sectors, being the sectors with the highest recurrence of no measures taken, as most of the Swedish responses came from those two sectors. In Ukraine, in addition to the 2022 drought being milder than in the previous two years, it could be speculated that the war and consequent prioritization needs have hampered the capacity to respond.

Responses addressing the classification based on the EC Blueprint (Sec. 1.3) showed that responders mostly employed
'demand reduction' and 'supply increase' measures (respectively 13% and 14%), while efficiency and prioritization measures remained underused (6.3%) (Fig. 7). Respondents in France (40%), Spain (37%), and the UK (35.7%) showed the largest adoption of demand reduction measures, whereas the Netherlands (43%), the UK (28.6%), Hungary (27%), and Italy (23%) showed the largest use of supply-side measures. Prioritization and efficiency measures remained underused except in Italy where they constituted 19% of the responses. This contrasts with the advice by the EC Communication, which clearly states
the need to prioritize demand reduction measures, followed by improving efficiency, and only as a last measure, increasing supply.





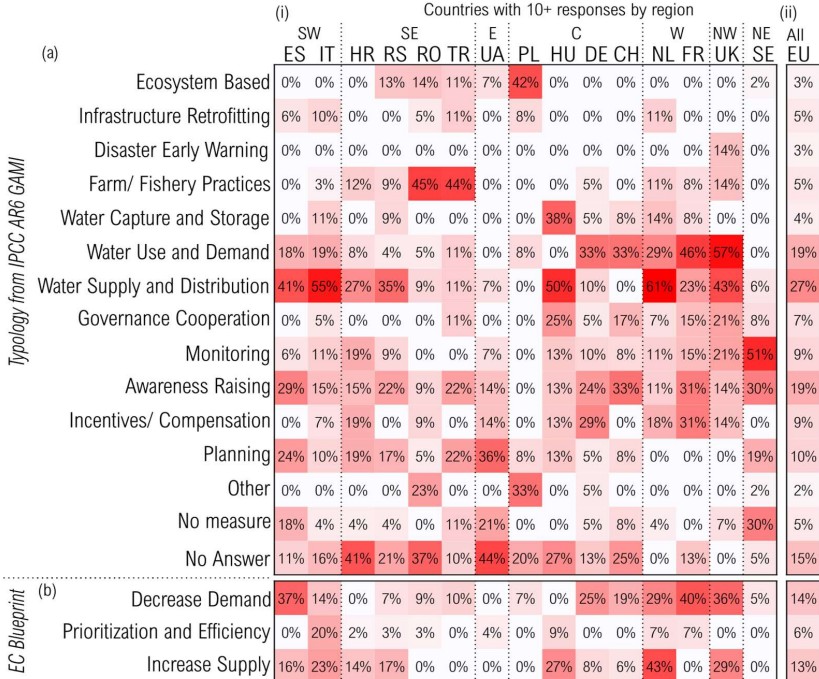

**Figure 7:.** Measures taken by the respondents organised by country (only countries with 10+ responses), and region. The numbers represent the percentage of respondents using a specific measure as relative to the total number of respondents for that country. Depending on the response, multiple measures could be identified for the same response. The European regions are described using the acronyms: SE (southeast), SW (southwest), E (east), C (central, W (west), N (Northwest), NE (northeast). Countries are indicated using their two-letter country code.

### 3.3.2 Perceived effectiveness of the response

Respondents were asked to rate the effectiveness of the measures taken during the 2022 drought on a scale from 1 (not effective) to 5 (very effective). The key features of results are depicted in Fig. 8. 25% of respondents rated the effectiveness of their measures between 1 and 3, meaning non-to moderately effective. Conversely, 16 % rated the measures as effective (4), and 9% very effective (5). Additionally, 11% were unable to answer, 11% responded "I don't know," and 16% marked "not relevant". Sweden and Ukraine in particular show a high rate of non-valid answers, possibly reflecting the fewer and less severe impact of the 2022 drought for the former, and the effects of the war for the latter. NGOs generally rated their efforts as least effective in managing drought risk, with 57% of the responders giving a rating of 1 to 3. Scientific organizations also reported below-average effectiveness (18% 1 to 3, but with only 32% valid responses) and a higher occurrence of "not relevant" responses (26%), reflecting their indirect role in drought response. Both private and public/governmental organizations reported effectiveness levels close to the European average, though only 27% of public/governmental organizations considered



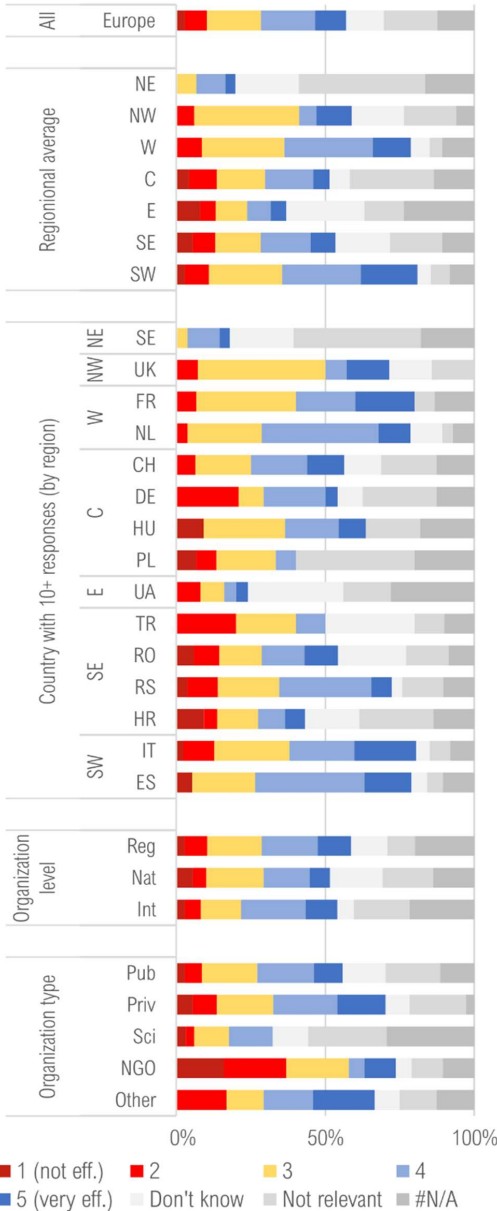

their measures effective or very effective. Organizations operating at the international level were the most positive on the effect, with 32% rating measures as 4 or 5. Regional-level organizations followed (30%), with national-level organizations being less certain about the effect (22%). Despite these variations, the differences across organizational levels were overall small and aligned closely with the overall assessment at the European level when excluding non-valid answers. The countries with the highest share of respondents indicating measures taken to be effective or very effective were Spain (53%), the Netherlands (50%), and Italy (42%). On the other hand, the highest share of effectiveness rated between 1 and 3, was found in the UK (50%), France (40%), and Italy (38%). A more detailed overview of the findings is available in the supplement (Table S3).

A notable share of respondents (18%) selected "not relevant" for their measures. This option was originally intended for organizations whose actions, like monitoring and data collection, do not directly impact drought management or for those who took no measures. To avoid misinterpretation (inferring 'not relevant' as 'not effective at all'), the option (less than 1) was placed separately from the 1-to-5 scale in the questionnaire. The responses indicate that 26% took no measures, and 24.1% did not answer the question. Among those who took measures, the most common were monitoring (17%) and awareness raising (9.5%). This suggests respondents correctly interpreted the question, though some misinterpretation might have occurred, potentially overestimating the reported effectiveness (Fig. S2 in the supplement). Swedish respondents accounted for 31% of "not relevant" responses, reflecting their fewer and less severe drought impacts and focus on monitoring.

**Figure 8:.** Effectiveness of the measures taken by the respondent's organization (Question 14 in the survey). The options included values 1 to 5 on a scale from "not effective" to "very effective" respectively,



"I don't know" indicating the respondents lacked knowledge on the effects of the measures that their organization put in place, "not relevant" meaning that the measures taken are not meant to impact drought response (e.g. scientific organizations collecting data), and the option to leave the question blank if the respondent was not able to answer the question for unspecified reasons. The values displayed are the percentage of valid answers for each sub-group. The first row shows the values for all the responses received and is labelled as "All / Europe". The sub-groups presented are individual countries with at least 10 answers grouped by region (see Sec. 2.1.3); and regional averages (including countries with less than 10 answers). The European regions are described using the acronyms: SE (southeast), SW (southwest), E (east), C
(central, W (west), N (Northwest), NE (northeast). Countries are indicated using their two-letter country code.

### 3.3.3 Changes in drought risk management

According to the survey, 79% of the respondents across the whole of Europe considered that drought risk management will become more significant for their organisation. In contrast, only 9.6% consider the opposite. Drought is expected to become a more significant risk to manage for all sectors, as indicated by the responses given for Freshwater Aquaculture and Fisheries
(88%), Public Water Supply (86%), and Waterborne Transportation (85%). Again, water managers operating on ecosystems are among the most certain that drought risk management will become more relevant, with Terrestrial Ecosystems, Freshwater Ecosystems, and Wildfires reporting 88%, 87%, and 87%, respectively. Switzerland (27%), Romania (20%), and the Netherlands (18%) are the countries where the perception that drought risk management will not become more important for the organization represent the highest – although rather low - percentages. Conversely, respondents from France (100%), Spain
(95%), the UK (93%) and Italy (91%) indicate that drought management will become more important.

Comparing the management of the 2022 drought with that of the 2018 drought, most organisations noticed increased drought awareness after the 2018-2019 drought, but not all could translate this into increased preparedness or a more effective response (Fig. 9). It is worth noting that there is a rather high correlation between 'more awareness' and 'more preparedness' or 'effective in the response' (correlation coefficient of 0.52 and 0.46 respectively). Among the organisations reporting increased awareness,
49% reported increased preparedness, and 46% reported increased effectiveness (40% reported both). There is an even stronger correlation (correlation coefficient of 0.7) between 'improved preparedness' and 'improved effectiveness' in the responses. Only a few organisations were less aware/prepared/effective in 2022 than during the 2018-2019 drought event (respectively, 3%, 5%, %) (Table S5 in the supplement presents these results in more detail).

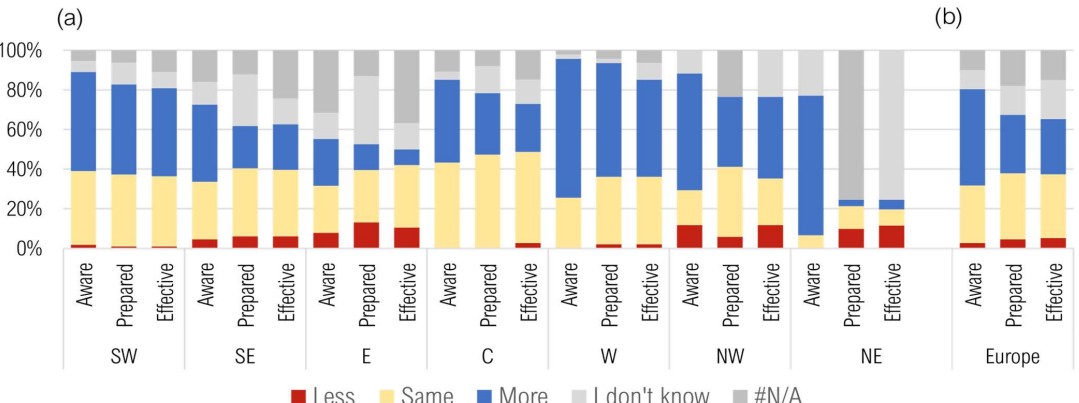



**Figure 9:** Changes in awareness, preparedness, and effectiveness in the response between the 2018 and 2022 drought events, according to the respondents. The respondents could answer "more", "same", or "less" to the three questions "How aware/prepared/effective was your organization in 2022 compared to 2018?". The option "I don't know" and the possibility to leave the question blank (i.e. "#N/A") were also available. The results are presented at the European level (i.e. all responses), and at the regional level. The European regions are described using the acronyms: SE (southeast), SW (southwest), E (east), C (central, W (west), N (Northwest), NE (northeast). Countries are indicated
using their two-letter country code.

### 3.4 Regional Spotlights

### 3.4.1 Catalonia

In 2022, most of Spain experienced a severe drought. The SPEI-12 for December 2022 reached values lower than -2.3 in the north of the country, only slightly less severe than the value of -2.9 reached in the 2004-2008 drought (Kreibich et al., 2023).

Out of the 19 respondents in Spain, representing various sectors, six indicated that the impacts of the drought were noticeable even before March 2022 (Fig. 5, Sec. 3.2). This difference in perceptions among respondents can be attributed to the fact that drought began in some regions of Spain already in June 2021, as shown in Fig. 4. Furthermore, most Spanish respondents reported that the drought persisted beyond September 2022, indicating its long-lasting nature (Fig. 5, Sec. 3.2). The meteorological drought, as identified by a SPEI-12 below -1, started in June 2021 in most parts of Catalonia and is still ongoing

in 2024 (GenCat, 2024).

Water resources in some areas of Spain have been dwindling due to long periods of meteorological drought. This situation is not uncommon, but it has worsened in recent years due to an increase in evaporative demand caused by global warming. To adapt to the current situation, there are 1225 dams in Spain, out of which 372 are categorized as high storage dams, with a combined storage capacity of 56,000 hm³ of water. These multipurpose dams, used primarily for irrigation, hydropower and

drinking water supply, are also utilized for flood control and ecological discharge maintenance in some cases. During wet years, hydropower production in Spain exceeds 40,000 GWh, whereas during dry years, it drops to less than 25,000 GWh. The average of recent years is 32,500 GWh, which represents 17% of annual energy production. Spain has the largest irrigated area in Europe, with 54% using localized irrigation systems. Despite irrigation occupying only 23% of the cultivated area, it contributes to 65% of the final agricultural production. During a drought, there can be a conflict of interest between the different

dam objectives (Ward et al., 2020). Most aquifers in Spain are over-exploited, and the legislation considers the state of the reserves in the dams as a criterion to distinguish the different drought alert levels.

In the case of Catalonia, the persistent drought has led to a SPI-12 index below -3, and surface reservoirs below 16% of their capacity as of February 2024, over two years after the start of the drought. Between 2022 and 2024, the Government has decreed several "emergency situations" due to hydrological drought in more than 230 municipalities across Spain, including

the Barcelona Metropolitan Area, affecting more than six million people. This involves restrictions to general water uses, including urban water use. Among the measures taken, it is possible to find a progressive restriction on domestic water consumption down to 160 liters per person and day. There are also restrictions for other economic sectors, with reductions of 80% for agriculture irrigation, or transition to reclaimed 50% for livestock and 25% for industries. Other measures involve the reduction of the ecological river flows established from the Sectoral Maintenance Flow Plan (e.g., Muga from 1,200 to 40 l/s,



Ter from 5,500 to 600 l/s, and Llobregat from 4,300 to 250 l/s), and a moratorium on the start-up of economic activities that require intensive use of water. Fines have been imposed on towns that fail to comply with the decree of maximum consumption. Additionally, there has been the announced need to rethink the water fee to penalize significant losses in the supply network or large consumers, and the request to close showers and pools in some sports and touristic facilities. Measures such as bringing water in ship carriers from other parts of Spain, which was implemented during the drought in 2008, have also been proposed.

However, the transfer of water from one river basin to another remains a sensitive topic. Under this crisis scenario, which is the worst drought registered in Catalonia, around three quarters of the water consumed in Catalonia no longer comes from dams, but from a combination of groundwater resources, regenerated water and desalination plants. Spain's aquifers are capable of storing up to 400,000 hm3 of water, which is seven times the capacity of the dams.

**3.4.2 Italy**

In Italy, the impacts of the 2022 drought were observed as early as March (depending on the sector), and persisted until after September 2022 (Fig. 5). The damage was severe across social and economic sectors, particularly for the Public Water Supply, Energy, Fisheries and Agriculture sectors (Fig. 6). The entire peninsula grappled with significant water scarcity (Fig. 2), distinguishing the 2022 drought as more severe than the one experienced in 2018-2019. Amid ongoing water scarcity, the Italian government responded with Decree-Law Drought No. 39 of 2023 (C.d.D., 2023), emphasizing urgent provisions to

counter water scarcity and enhance water infrastructure. Mitigation measures included simplifying water infrastructure procedures, increasing reservoir volumes, rainwater harvesting for irrigation, treated wastewater reuse, and desalination projects. A dedicated committee assessed projects and the National Commissioner for Water Scarcity expedited interventions. Regional and public administrations played a pivotal role in implementing drought management strategies, with a primary focus on Public Water Supply, Agriculture, and Water Quality. During the first half of 2022, approximately 60% of these

organizations took proactive measures to address the impact of the drought. Respondents prioritized Public Water Supply, Agriculture, and Livestock Farming (Fig. 6). Most organizations (61%) initiated drought risk management measures during the first half of 2022. Still, despite legislative efforts, questionnaire responses indicated gaps in drought preparedness. Only 28% had both short- and long-term drought management plans, and 51% reported a complete lack of plans.

    By far, the most commonly implemented measure regarded water distribution management, with Water Supply and

Distribution being the most common (55%), followed by Water Supply and Demand (19%) (Fig. 7). This demonstrates a tendency for water managers to guarantee business-as-usual operations in times of water scarcity. As exemplified by this statement provided by one of the respondents from Italy when asked about how drought risk management is changing in their organization: "Greater control and assessment of the situation through monitoring, elimination of [water] leaks or waste, exploration of new sources, and implementation of new storage facilities". Still, respondents have also shown awareness of

the need to reduce demand: "It is necessary for the authorities to allow extraordinary works and permits to prevent the loss of well zones. Even the sole reduction of withdrawals during hot periods along the riverbank would be a response". Other measures included: awareness rising (15%), water capture and storage (11%), monitoring (11%), infrastructure retrofitting



(10%), and planning (10%). Despite challenges, respondents emphasized the relevance of ongoing efforts to enhance water resilience. Yet, the Italian context shows a strong preference towards supply side measures, emphasizing the need to meet
water demand even during drought periods.

## 4    Discussion

4.1. Challenges of drought risk management

### 4.1.1 Increasing drought risk

One notable consensus among the responders of the survey is the recognition that drought is increasingly becoming a more
significant risk across Europe. They anticipate that their respective organizations will place drought risk management at higher priority in the future. This corroborates the increase in frequency and intensity of drought hazard presented both in this and in previous studies (Markonis et al., 2021; Moravec et al., 2021, Spinoni et al., 2018, Ionita et al., 2022, Jaguus et al., 2021, Semenova & Vicente-Serrano, 2024).

Beyond the higher frequency, this study highlights the extensive scale of drought impacts, prompting drought risk management
measures across all European countries. This underscores the potential benefits of continent-wide coordination already highlighted by previous research (Blauhut et al., 2021; Hervás-Gámez & Delgado-Ramos, 2019; Publications Office of the European Union, 2023; Rossi, 2009; Stein et al., 2016). This shared understanding of growing drought risk and the increasing need for drought risk management emphasizes the continent-wide scale of the challenge and further reinforces the need for collaborative initiatives and unified guidance. Our findings align with extensive research showing how droughts transcend
national borders and emerge as cross-boundary challenges (Herrera-Estrada et al., 2019), impacting the entire European continent (Ionita et al., 2022; Rakovec et al., 2022; Schumacher et al., 2024; Spinoni et al., 2018; Toreti, et al., 2022), and requiring European-level direction in drought risk management and response (Blauhut et al., 2022; Hagenlocher et al., 2023; Stein et al., 2016; van Daalen et al., 2022).

The study stresses the need to assess water prioritization criteria, considering the actual impacts on various sectors and
adjusting the allocation strategy to ensure a more equitable and effective distribution of water resources. The example of Catalonia only highlights the challenges of managing water use prioritization between sectors.  Yet, observed or expected impact should not be the only indicator of prioritization as it is in itself influenced by prioritization and other measures. Sectors of key importance for human and environmental well-being (e.g. Public Water Supply and Aquatic Ecosystems) must be prioritized regardless of impact due to their importance in the functioning of the SES (Rossi et al., 2023).

### 4.1.2 Spatial and temporal evolution of drought

Droughts are long-lasting events that can span over several seasons. Their impact can affect different aspects of SES depending on the response time of the system in question (e.g. depletion of water supplies can last for years, while the forestry sector



might only show visible effects years after the drought "event", and governance effects may take years to materialize). This characteristic of drought as a complex crisis with long-term systemic ramifications is explored under the notion of "drought as a continuum" by Van Loon et al. (2024).

Respondents indicated that the impact of the 2022 drought extended beyond the observation period covered by the questionnaire in 44% of cases (i.e. after September 2022), and spanned the entire observation period in 7% of all cases (i.e. from before March 2022 to after September 2022). This is exemplified in the autonomous region of Catalonia (ES), where the drought is ongoing as of 2024. The Catalan case is a clear example of prolonged drought impacting different components of the wider system over time, from the hydrological, ecological, and socio-economic systems.

Still, drought risk management in Europe generally defines drought as an extraordinary, time-confined, event with a predominant seasonal occurrence (Stein et al., 2016). As a consequence, monitoring and drought management teams are typically assembled on a seasonal basis and are disbanded once the crisis has subsided (this differs across European countries), with consequent overlooking of long-lasting and lingering impacts of drought. Additionally, a crisis approach to drought risk management frames drought as a crisis and justifies extreme measures that can have long-lasting consequences. This is exemplified by the Italian case, where an ad hoc drought commission was instituted to tackle the crisis.

Most European member states present some version of article 4(6) of the WFD in their national water basin management plans, allowing them to reduce or forego environmental outputs during times of drought (Publications Office of the European Union, 2023). This overlooks the complex nature of drought risk management and the ramifications that short-sighted measures can have. Instead, research shows that a systemic risk perspective is necessary to manage complex crises like droughts (Wilhite et al. 2019; Hagenlocher et al., 2023), and that European-level drought risk management should strive to implement it (Stein et al., 2016). Drought hazard and impact monitoring and forecasting should be strategic efforts that not only take into account the physical aspects of drought, but focus on water scarcity and its relation with impacts (Sutanto et al., 2019; Shyrokaya et al., 2024). A systemic perspective, in this instance, is necessary to show how drought impacts can be worsened by decisions taken during "normal" times (Hagenlocher et al., 2023; Kallis, 2008; Wilhite et al. 2019). Moreover, for the principles stated in the EC communications about drought and water scarcity to be effectively implemented, this systemic perspective must prioritize holistic measures that account for environmental conservation and water use reduction across all sectors and users.





**4.1.3 Drought risk management measures**

The measures taken by organizations predominantly focus on immediate operational concerns, such as water supply management, to ensure business continuity during droughts. In particular, supply-side measures were the most commonly used, especially in countries where agriculture plays an important economic role (e.g. Italy, Spain, the Netherlands). This is in direct contrast with the recommendations from the EC communications *Blueprint to Safeguard Europe's Water Resources* and the *Water Scarcity and Drought Policy*, which instead stress the importance of prioritizing demand reduction and improving efficiency

*"The main issue was high demand rather than supply shortfall - the distribution network encountered issues due to the high demand in May-July and eased off in August. All sources were utilising their peak output for 2-3months whilst planned outages were postponed."*

UK Responder (Question 12)

before opting for increasing supply (Stein et al., 2016). Additionally, both recommendations and research stress the importance of reducing water use in general. Simply providing more water surplus by either increasing supply or improving efficiency leads to an increased water-use, which quickly nullifies the surplus, as demonstrated by the reservoir effect (Di Baldassarre et al., 2017, 2018). Instead, by prioritizing water demand reduction and increasing efficiency, organizations should reduce water consumption, and avoid maladaptive practices and path-dependencies.

As the Italian case exemplifies, short-term and supply-side measures are likely favoured as they address immediate concerns of the responders and sectors involved (Teutschbein et al., 2023). This is an example of "salience-bias", where disproportionate weight is given to more immediate concerns due to proximity, memory, perspective, or deliberate choice, potentially leading to suboptimal decisions (Bordalo et al., 2020; Garcia et al., 2020; Garcia & Islam, 2021). Still, as the "hydro-illogical cycle" shows, it is challenging to mainstream drought risk management measures during periods of non-drought, depriving preparedness and mitigation measures of their effectiveness, making response measures more necessary (Wilhite et al., 2005). This situation can be further exacerbated by development policies that are not aligned with drought management policies and instead negatively impact them (Kallis, 2008). Rather, drought risk management should embrace an integrated and systemic approach, as proposed by IDM, by avoiding short-term measures when these are shown to be less effective than proactive, long-term, and systemic ones (Wilhite et al. 2019; Wendt et al. 2021).

The allocation of water resources during droughts presents a complex challenge, particularly in balancing the needs of highly impacted sectors against those less severely impacted, or where the risk seems more imminent. For instance, from the responses gathered in this study, it emerged that sectors such as Forestry in Germany and Energy, Industry, and Fisheries in Italy received a low priority for water allocation despite experiencing significant drought-related impacts. Conversely, sectors such as Public Water Supply, Water Quality, Water Transportation, and Tourism were considered high priority across multiple countries, even if their impact seems less severe. However, it is crucial to acknowledge that the perception of milder impacts in certain sectors may also be influenced by the effect of prioritization itself. Moreover, sectors like Forestry, where the full extent of impact may take longer to manifest (Shyrokaya et al., 2023), might receive lower priority despite their critical importance,



especially with significant increase of drought impacts in the forestry sector (Rossi et al., 2023). This prioritization discrepancy underscores the need for a nuanced approach to manage the diversity of drought impact across operational levels and sectors
in response to varying degrees of drought severity. Consequently, decision-makers must strike a delicate balance between addressing immediate needs and ensuring equitable resource allocation across sectors, especially considering the potential long-term consequences of drought impacts.

This study shows that during the 2022 European drought, water managers expressed a lack of emphasis on longer-term adaptive measures; as also highlighted by the Italian case. This is supported by research showing that despite the WFD supporting
adaptive water management approaches, implementation generally follows a standard responsive approach as institutional practices, competencies, and skills are not aligned to what an adaptive approach would require (Voulvoulis et al., 2017). This suggests a potential gap in strategies, with an opportunity for organizations to consider more sustainable and forward-looking approaches to drought risk management, such as ecosystem-based adaptation (IPCC, 2022a). This is also highlighted by the overall preference for sub-seasonal forecasting and short-term drought management plans over seasonal forecasting and long-
term plans (Biella et al., 2024b). Yet, research warns against the risk of maladaptation that reliance on short-term information alone can cause (Biella et al., 2024a). Consequently, a systemic and long-term perspective in the DMPs focusing on demand-reduction can be instrumental in avoiding maladaptive outcomes and path-dependencies (Hagenlocher et al., 2023).

### 4.1.4 Shifts in drought risk management

We find clear regional and country-level differences in drought risk management across Europe, likely reflecting the varying
impacts of drought in the region and scale considered. These differences in drought risk management can be observed across all aspects, from the type of measures taken, to the effectiveness of these measures, to the reported changes in drought risk awareness and preparedness. Due to the limited sample size of some of the countries (see Table S1 in the supplement), and the high rate of "no answers" in some of the categories, it is not possible to draw a generalizable conclusion for all sub-groups. However, the consistency reported across different aspects of drought risk management should be taken as strong evidence of
the large differences currently present in European drought risk management. Similar differences have been highlighted in the drought preparedness of water managers across European countries (Biella et al., 2024b). This also is supported by reports showing the diverse drought risk governance landscape of the continent (Publications Office of the European Union, 2023). This discrepancy in drought risk management capacity across European countries emphasizes the urgent need for continent-level guidance, acknowledging the diverse challenges faced by different regions. Despite the various EC communications on
droughts and the inclusion of drought in many strategies, the lack of a unified policy with binding force means that the drought risk management landscape of the continent remains diverse (Stein et al., 2016). Factors, such as availability of resources and drought risk awareness, likely contribute to the disparities in drought risk management capacity observed. The development of a *European Drought Directive*, would be instrumental in levelling out the difference among countries (Blauhut et al., 2022). The survey results point to a trend where organizations are becoming more conscious of the risks posed by drought and suggest
that time is ripe to mainstream drought risk management into policy in Europe. While awareness of drought risk increases



across Europe, preparedness and effectiveness are lagging behind. The survey demonstrates clear differences at the regional level, with respondents from Eastern, South-Eastern, and Northern Europe displaying minor changes in drought risk management compared to their counterparts in South-Western, Central, and Western Europe. Research shows that mainstreaming drought risk management is most effective after times of crisis, when awareness is high (Cavalcante et al 2023,
Kreibich et al 2023). This is evident in the EU, as several countries with drought legislation in place have promoted it following the large drought of the last decade (Publications Office of the European Union, 2023; Bartholomeus et al., 2023). Still, EU-level policy mainstreaming is often a complex, lengthy, and highly political process and compromising (Deters, 2018; Kaika, 2003). The results of this study clearly show that European water managers display high levels of drought risk awareness, while preparedness still has room for improvement. This means that it is essential to take advantage of this mainstreaming
window to promote drought risk management policy across Europe. It is the role of research to ensure that awareness remains high in times of non-crisis, avoiding the hydro-illogical cycle.

## 5 Recommendations for European drought risk governance

### 5.1 Gaps in European drought governance

This research underscores the necessity for cohesive, European-wide coordination in addressing the increasing drought risk,
the scale of the threat posed by drought, and the interconnectedness and co-dependence of ecosystems and socio-economic sectors across the continent. The regional differences and the differences in the adaptive pathways across countries show the need for a coordinated approach to address shared vulnerabilities, foster collaboration and coordination, and increase equity (EC et al., 2015; Hagenlocher et al., 2023; Publications Office of the European Union, 2023; Stein et al., 2016).

Nevertheless, the EU lacks a unified drought policy, and the reliance on a framework of other water-related directives and
non-binding communications limits this progress (Hervás-Gámez & Delgado-Ramos, 2019; Publications Office of the European Union, 2023; Stein et al., 2016). The 2000 WFD remains the only existing binding directive loosely dealing with drought; yet it does not specifically address it nor defines it, only mentioning drought together with floods (Publications Office of the European Union, 2023; Stein et al., 2016). Furthermore, the WFD's framing of droughts (and floods) as "force majeure" can justify non-compliance with environmental needs (DIRECTIVE 2000/60/EC). This is in contrast with a vast body of
research showing that viewing droughts as exceptional events overlooks their lasting and systemic impacts and increased risk (Hagenlocher et al., 2023; Van Loon et al., 2024; Walker et al., 2024; Markonis et al., 2021; Moravec et al., 2021; Spinoni et al., 2018; Ionita et al., 2022). Still, the WFD offers a solid base on which European drought risk management can be developed, as it crucially defines catchment-level water management, environmental output requirements, unified monitoring, and international collaboration for transboundary basins (Publications Office of the European Union, 2023; Stein et al., 2016). The
catchment-centred perspective (instead of administrative borders) in particular suits the need of cross-country drought risk management. Finally, its cross-sectorial focus and the adaptability of its 6-year revision cycles align with the needs of a systemic drought risk management approach.



Following the WFD, the 2007 EC communication on water scarcity and droughts and the 2012 Blueprint to Safeguard Europe's Water Resources (also an EC Communication) have also been instrumental in defining DMPs, and promoting country-level drought risk management through a clear emphasis on the importance of water conservations measures (Hervás-Gámez & Delgado-Ramos, 2019; Stein et al., 2016). However, despite their ambitious principles, the EC Communications of 2007 and 2012 remain non-binding, crucially lacking mandatory power over EU member states' legislation, as well as diverse, binding policy options (Stein et al., 2016).

Droughts are recognized a priority in other EU policy frameworks dealing with specific sectorial issues; adding to the need of cross-sectoral policy. The *European Green Deal* and the 2021 *EU Strategy on Adaptation to Climate Change* have a dedicated *Group on Water Scarcity and Drought* in the 2022-2024 *Programme for the Common Implementation Strategy for the Water Framework and Floods Directives*. Other relevant directives include the EC *Flood Directive* (2007), *Groundwater Directive* (2006), and *Habitats Directive* (1992). Moreover, the EU's *Common Agriculture Policies* (CAP), a vast framework governing agriculture since the 1950s, also defines tools for drought governance (Stein et al., 2016). However, these directives only deal with drought within the boundaries of the sectors that they address. For example, although CAP includes many ecosystem-focused principles, it also includes stabilization mechanisms that might encourage risky agricultural practices during droughts, which clearly indicates/points to a lack of systematic/holistic perspective (Stein et al., 2016). Similarly, measures in flood risk management and reservoir management can indirectly affect drought risk management. Consequently, without unified guidance taking a systemic, sustainable, and long-term approach drought risk management strategies may risk incurring in maladaptation, especially when competing with the economic development interest of other sectors.

### 5.2 A way forward: The European Drought Directive

This study completes a series of research efforts highlighting the need to establish European coordination and guidance on drought risk management (Blauhut et al. 2021, Moravec et al. 2021, Stein et al 2016, Rossi 2009, Hervás-Gamez & Delgado-Ramos 2019, European Drought Atlas 2023). Supporting the recommendations by Blauhut et al., (2022), we advocate for the development of an *EU Drought Directive*. While the EC communications on drought (namely, the WS&D and the Blueprint) already present many ambitious principles, a legally-binding directive is necessary to ensure their implementation and create consistency among different countries. This *EU Drought Directive* should establish principles of drought risk management, provide coordination, and guidance at the EU level, and set up cooperation agreements with third countries of interest (e.g. Switzerland, the UK, Norway, Ukraine, and countries in the western Balkans). At the same time, implementation should be carried out at the member state level, being tailored to the local context and operational needs. This approach is similar to that already provided in the *Floods Directive* (Directive 2007/60/EC). Additionally, we suggest amending the WFD to include clear drought risk management principles as a necessary first step, as the framework already introduces valid water resource management principles that can be effectively applied to drought risk management (e.g. catchment-based management, and international coordination guidance). The WFD is can also provide the ideal governance framework for a holistic and integrated approach which manages both drought and flood risk. We believe a European *Drought Directive* should:



1. *Define the principles that guide drought risk management.* These have already been indicated in the non-binding EC Communications, have counterparts in the Flood Directive, or have been defined by research. These principles are:

   a. *Managing drought risk, not drought hazards.* While periods with less precipitation cannot be prevented it is possible to reduce their adverse impacts on human health, the environment, and socio-economic activities. A risk approach to drought risk management requires considering all aspects of risk and not focussing on the hazard alone.

   b. *Drought is a continuum.* Droughts are not entirely exceptional events. They occur with relative frequencies, and their impacts propagate through the socio-economic system. Hence, drought risk management should not merely be responsive, seasonal, and with a crisis-based approach. Instead, it should adopt systemic, integrated, and long-term risk management perspectives that address water scarcity and stresses even during non-drought periods. This approach helps avoid path-dependency, lock-ins, and maladaptation.

   c. *Environment-centred drought risk management.* Environmental needs should be prioritized also during drought periods, especially in case of long-term damaging impacts on the ecosystem. This means that drought should not constitute a valid reason to forego environmental needs in favour of economic activity. Instead, drought risk management should ensure and protect the ecosystem's capacity to support natural and human activity (ecosystem services).

   d. *First reduce demand, second improve efficiency, last increase supply.* The measures aimed at managing drought risk need to prioritize water demand reduction, and reduction of dependencies. A second priority is to increase water use efficiency in the system. Yet this increased efficiency should come hand in hand with demand reduction. Lastly, supply increase measures and infrastructural measures should only be considered where the first two options are not feasible. Maladaptive outcomes, such as increased water dependence and the reservoir effect, should be avoided.

2. *Provide guidance and coordination for drought risk management.*

   a. *Provide guidelines for the definition of drought.* The directive contains a general definition of drought, while allowing Member States to tailor the definition to their contexts. This requires including indices representing different types of drought (meteorological, soil moisture/agriculture, hydrological droughts) in response to the wide range of drought impacts encountered.

   b. *Provide guidance for international coordination in drought risk management.* Drought risk management should be carried out on the principles of shared/transboundary    river basin as already defined in the WFD. To do so, amending the WFD to include drought is necessary. The Directive must also provide guidance on collaboration with countries that are not members of the EU due to the sectorial cross-border dependencies and shared river basins. The flood directive offers an example of such guidance.



    c. *Provide guidelines for the development and implementation of national drought risk management policies*
    *following the 10 steps process detailed in the National Drought Management Policy Guidelines: A Template*
*for Action (WMO & GWP, 2014)*
    d. *Provide deadlines for key steps in the development of national drought risk management policies:*
        i. *Carry out preliminary drought risk assessment.*
        ii. *Carry out drought risk assessment and draw drought risk maps.*
        iii. *Develop Drought Risk Management Plans at the national and regional level.*
840         iv. *Mandate the development of Drought Risk Management Plans for private actors in key sectors.*

## 6   Conclusion

The 2022 European drought, a continent-wide event, has exposed numerous deficiencies in the existing European water management framework. This study provides an overview of the 2022 European drought, highlighting the connections between its physical aspects (the hazard), the perceived sectoral impacts by water managers, and the drought risk management strategies
employed.

The study reveals that drought is increasingly recognized as a significant risk across Europe, with growing awareness, preparedness, and response capacity among institutions and organizations. As droughts are becoming more frequent and intense in the warming climate, the need for continent-wide coordination and data sharing is of utmost importance. Despite existing measures, droughts are often treated as extraordinary events, leading to short-sighted and potentially maladaptive
responses. The study highlights the importance of adopting a systemic, integrated, and long-term perspective in drought risk management at the continent level, prioritizing demand reduction and ecosystem health. A *European Drought Directive* is recommended to unify and enforce drought risk management policies at the national, regional and catchment scales, ensuring coordinated efforts across the continent. This directive should guide the development of drought management plans, emphasize risk management over crisis response, and prioritize environmental outputs and water demand reduction. Coordinated
European-level action is essential to address the shared vulnerabilities and complex nature of droughts, ensuring effective and sustainable management of this escalating risk on the climate resilient pathway for all European countries.

**Competing interests**

At least one of the (co-)authors is a member of the editorial board of Natural Hazards and Earth System Sciences Journal.

**Code Availability**

All codes used for the statistical analysis can be made available upon individual request.



**Data availability**

The data collected during the survey contains information that might allow to identify some of the respondents. Hence, all data collected through the survey has been stored on DitA's workspace and can be made available upon request. Climate-related data is freely available as described in Sec. 2.1.

**Interactive computing environment**

No interactive computer environment is available.

**Sample availability**

No physical samples were collected.

**Video supplement**

No video supplement was developed.

**Supplement link**

The link to the supplement will be included by Copernicus, if applicable.

**Author contribution**

*Conceptualization:* The conceptualization of the article involved a large group of authors as the initial idea was developed
during the *Drought in the Anthropocene* annual workshop in Uppsala in July 2022 and was defined during a first online meeting in October the same year. All the following authors were involved in the conceptualization of this manuscript as they were present and actively participated during either of those events: AM, AS, AT, AvL, BM, CT, DC, ER, ES, FR, FT, GDB, IP, J-PV, LMT, LB, MW, MI, PT, RH, RB, SS, SH, SC, SM, SJB, SK and VN. *Methodology and Data Collection:* The following authors were involved in the designing, translation, and dissemination of the survey: AM, AS, AT, AvL, BM, CT, DC, ER,
ES, FR, IP, MCL, MMdB, ML, MW, MI, PT, PA, RH, RV, RB, SS, SC, SJB and VN. Additionally, the following authors were involved in the collection and handling of climate data: IP, MI, PA, RH and SS. *Project Administration:* AS, MI, and RB were responsible of management and coordination of the team's research activities throughout the development of the study. *Supervision:* GDB and LMT offered invaluable supervision at various stages of the development of the manuscript. *Visualization:* The figures, tables and maps present in the manuscript were created by: AS, MI, PA, RB and SS. *Writing:* The
original draft was mostly prepared by a core team composed by: AS, LMT, MI, RV and RB. Additionally, other authors were



involved in writing specific sections of the manuscript: AM, BM, DC, ER, FR, IP, MCL and SC. Other authors were involved in the reviewing and editing process, offering commentary and suggestions to the original draft: AT, CT, DW, FT, MMdB, ML, MW, PA, SS, SJMG and SK.

**Special issue statement**

The statement on a corresponding special issue will be included by Copernicus, if applicable.

**Acknowledgements**

*The research work was partly funded by:*

European Union's Horizon 2020 research and innovation programme under the Grant Agreement Number 101037293: ICISK Innovating Climate services through Integrating Scientific and local Knowledge.; European Union's Horizon 2020 research
and innovation programme under the Marie Sklodowska-Curie Grant Agreement No. 956396 (EDIPI Project); partially supported by a grant of the Ministry of Research, Innovation and Digitization, under the "Romania's National Recovery and Resilience Plan - Founded by EU – Next Generation EU" program, project "Compound Extreme events from a long-term perspective and their impact on forest growth dynamics (CExForD)" number760074/23.05.2023, code 287/30.11.2022; The Wageningen Data Driven Discoveries in Changing Climate (D3-C2); National Hydrological Monitoring Programme,
supported by the Natural Environment Research Council award number NE/R016429/1 as part of the UK-SCAPE programme delivering National Capability; NERC project 'IndicatoRs to Impacts for drought Surveillance and management' (IRIS, Grant Number: NE/X012727/1) by the European Union (ERC, PerfectSTORM, ERC-2020-StG 948601); European Union's Horizon Europe research and innovation programme under the Grant Agreement Number 101121192: MedEWSa - Mediterranean and pan-European forecast and Early Warning System against natural hazards, and from the European Union's Horizon 2020
research and innovation programme under the Grant Agreement Number 101003876: CLINT - Climate Intelligence: Extreme events detection, attribution and adaptation design using machine learning; .RETURN Extended Partnership and received funding from the European Union Next-Generation EU (National Recovery and Resilience Plan – NRRP, Mission 4, Component 2, Investment 1.3 – D.D. 1243 2/8/2022, PE0000005 – Spoke DS8); European Union's Horizon Europe research and innovation programme under the Grant Agreement Number 101003469-XAIDA; European Union's Horizon 2020 research
and innovation programme under the Grant Agreement Number 820712-RECEIPTSwedish Research Council for Environment, Agricultural Sciences and Spatial Planning (FORMAS, contract number: 942-2015-1123); Views and opinions expressed are however those of the authors only and do not necessarily reflect those of the European Union or the European Research Council Executive Agency. Neither the European Union nor the granting authority can be held responsible for the European Union's Horizon 2020 research and innovation programme.



*Finally; we thank the following people for their help in disseminating and translating the questionnaire, or for other types of*
      *support offered:*

Francesco Avanzi (CIMA foundation); Gregorio Pezzoli (UNIBG); Lotte Muller (VU Amsterdam); Irem Daloglu (Bogazici
University); Saeed Vazifehkhah (WMO); Shaun Harrigan (ECMWF); Florian Pappenberger (ECMWF); Conor Murphy
(Maynooth University); Mónika Lakatos (Hungarian Met Service); David W. Walker (WUR); Magdalena Smigaj (WUR);
Sina Khatami; Veit Blauhut (Freiburg University); Kevin Dubois (Uppsala University); Ferran López Martí (Uppsala
University); Gemma Coxon (University of Bristol); Ype van der Velde (VU Amsterdam); Niko Wanders (Utrecht University);
Matthijs ten Harkel (province Noord-Brabant); and Lars de Graaff (IVM-VU Amsterdam).

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
