# Peer review of "The 2022 Drought Needs to be a Turning Point for European Drought Risk Management"

_EGUsphere, 2024_

## Referee Comment (RC2)

This is my first review of the manuscript "**The 2022 Drought Needs to be a Turning Point for European Drought Risk Management**" by Riccardo Biella et al.

The paper addresses the increasing threat of drought across Europe, focusing in particular on the extreme 2022 event. The authors present a comprehensive analysis based on a large-scale survey of 481 water managers from 30 European countries, linking these responses with climate data to evaluate current drought risk management strategies. They find that responses remain largely reactive, with limited long-term or systemic planning, and advocate for a shift toward more proactive, integrated governance. Central to their argument is the call for a European Drought Directive, modeled after the existing Flood Directive, to harmonize risk management across the EU.

The manuscript is overall well written and provides timely, valuable insights into drought governance. While some of the findings—such as the reactive nature of current responses—may be expected, the strength of this paper lies in its empirical demonstration of these trends across a broad and diverse sample of practitioners. I believe the paper makes a significant contribution and is well suited for publication in this journal. However, I outline below several technical and methodological aspects that could be improved to enhance the manuscript's clarity, precision, and overall impact.

1. While the paper is overall informative and timely, I noticed **variability in the writing quality across sections**. Some parts are written in a highly polished and engaging style, while others would benefit from improved clarity, cohesion, and editorial refinement. This is understandable given the likely multi-author contributions, but I recommend a more thorough language review to ensure consistency throughout. Additionally, the manuscript is quite long and could benefit from tightening in structure and expression. Some sections—particularly methodological details or extensive contextual framing—could be more concise without sacrificing content. This would make the article more accessible and directive, improving its impact for both academic and policy audiences.

2. I recommend that the authors define **key terms** such as "drought risk" and "drought impact" etc.. early in the manuscript, preferably in the introduction (now reported after lines 240). These concepts are key to the study, yet their definitions and distinctions are currently scattered. A clear, upfront definition would improve readability and conceptual clarity, especially for interdisciplinary readers or those less familiar with drought governance literature.

3. I suggest including a summary **table** that outlines the key features of the EU-level directives and communications discussed in the paper—particularly the Water Framework Directive, the 2007 and 2012 EC Communications on Drought, and the Floods Directive. Such a table could include information on the directive's purpose, legal status (binding or not), scope (e.g. drought-specific or general water management), and its relevance or gaps in addressing drought risk. This would enhance clarity and reinforce the paper's argument for a dedicated Drought Directive.

4. In **Section 2.1** ("Climate data"), I suggest renaming the section to "Climate Data and Drought Assessment", as its current content focuses more on the definition and methodological framework of drought (e.g., SPEI, PET) than a traditional climate dataset description. This would better align reader expectations with the content presented.

   Furthermore, I recommend the authors:

   - Be more precise in the definition of drought. For example, include higher-than-normal temperatures alongside "abnormally low precipitation" to reflect common definitions and emphasize the role of temperature in increasing evapotranspiration (see line 211).

   - Include a threshold-based definition (e.g., SPEI < -1 for moderate drought) to help readers understand how drought severity is categorized.

   - Explicitly describe the CRU dataset used (temporal/spatial resolution, variables).

   - Justify the choice of the 1971-2000 reference period, especially in light of more recent climatological baselines.

   - List and cite the sources of additional variables (e.g., wind speed, radiation) used to compute PET with the Penman-Monteith method.

I acknowledge these might seem like technical details, but given that the paper targets a broad audience, including policy and decision-makers, being transparent and precise in this section is essential for replicability, credibility, and comprehension.

**5. Some methodological aspects**

At line 270, the authors acknowledge limitations due to uneven national responses, and thus restrict country-level analyses to those with more than 10 responses. However, this raises a broader concern: why is the analysis so heavily centered on countries, when drought is a transboundary risk that does not align with political borders?

I am not asking here to redo the analysis but I would encourage the authors to clarify the rationale for presenting both country-level and a large number of regional breakdowns—especially when many of these regions are effectively synonymous with a single country (e.g., SE = Greece + Cyprus). This risks redundancy and may obscure rather than illuminate regional trends. For me a simplified and more meaningful regional classification, such as Northern vs. Southern Europe, or humid vs. drought-prone zones, which would better reflect climatic, hydrological, and socio-institutional differences in drought exposure and response and improve readability and interpretability of the results as well as align better with the paper's framing of drought as a pan-European, systemic issue.

6. **Line 304-305**: the sentence is not clear.

7. **Line 320**. I suggest to cite here rather than at lines 79: *Avanzi, F., Munerol, F., Milelli, M., Gabellani, S., Massari, C., Girotto, M., ... & Ferraris, L. (2024). Winter snow deficit was a harbinger of summer 2022 socio-hydrologic drought in the Po Basin, Italy. Communications Earth & Environment, 5(1), 64.* This paper provides evidence that align well with the narrative in this part, especially regarding the role of snowpack as a hydrological buffer and its failure during the 2022 drought with its sociohydrological consequences.

8. **Lines 341**: the sentence is not clear.

9. **Section 3.4**: The paper includes detailed "regional spotlight" case studies on Italy and Catalonia, which are both valuable and illustrative. However, I recommend the authors explicitly clarify the criteria used to select these two regions for deeper analysis. Were they chosen due to data availability, impact severity, institutional diversity, or exemplary practices? Without this clarification, it may appear arbitrary, especially given that other regions also experienced significant drought impacts. Additionally, a brief mention of how these cases contrast with or represent broader patterns seen in the survey would help locate them more clearly within the Europe-wide analysis.

10. In **line 731**, the authors mention that disparities in drought risk management capacity are likely influenced by factors such as resource availability and drought awareness. I think it is important also acknowledging the role of governance systems, specifically the level of decentralization or federalism in water management. Countries with federal or devolved systems (e.g., Germany, Spain) may have highly region-specific approaches and coordination challenges, which can affect both preparedness and response. This for example what

happens in Italy in 2022. This institutional diversity is an important dimension of drought governance and could help explain some of the observed regional disparities.

11. **Figure 6** is visually dense and somewhat difficult to interpret effectively, especially given the number of countries, sectors, and metrics (impact severity and prioritization) displayed at once. I suggest the authors consider simplifying the figure, perhaps by initially presenting only panel (a) (impact severity), which already conveys the core message of regional differences quite well. Additional detail on prioritization could then be discussed in-text or moved to supplementary material if needed. Additionally, this figure highlights the challenges of the regional aggregation strategy. For example, in the SE region, Agriculture and Livestock appears as impact level 4 overall, but most individual countries in the group report level 5. Is this due to response imbalance (e.g., a high number of responses from Turkey relative to others)?

Based on the strengths of the manuscript and the relevance of its contributions, I recommend **acceptance after minor revisions**.

Christian Massari

---

## Author Response (AR1)

**Response to Reviewers**

**Manuscript Title:** The 2022 Drought Needs to be a Turning Point for European Drought Risk

Management

Authors: Riccardo Biella et al.

We are sincerely grateful to the Editor and the Reviewers for their time and thoughtful assessment of our manuscript. We highly appreciate the constructive and encouraging comments, which have greatly contributed to improving the clarity, structure, and potential impact of our study.

The reviewers' feedback helped us strengthen our framing, sharpen our argumentation, and improve both methodological transparency and editorial consistency. We have revised the manuscript accordingly, and we provide below a detailed point-by-point response. All reviewer comments are reproduced in italic, followed by our response in regular text. Where applicable, we have included line numbers referring to the revised manuscript with annotations.

**Reviewer 1**

We thank Reviewer 1 for their highly encouraging feedback and for acknowledging the scientific and practical relevance of our study. We particularly appreciate the recognition of the value of combining quantitative and qualitative data and the policy-oriented nature of our findings. We respond in detail below to the comments raised.

**General Comment**

This is an interesting paper, exploring a very important topic with applied research, bringing together quantitative and qualitative research on European drought management through the study of a recent drought in Europe (2022). The paper is well written and the content, findings, discussion and recommendations are of value to both academia and practitioners.

We are grateful for this positive assessment, which affirms the core objective of our work: to provide an empirically grounded, policy-relevant analysis of drought risk management based on the extreme 2022 drought event. We have revised the manuscript to further improve clarity, consistency, and policy relevance based on your detailed comments below. All references to changes in specific lines correspond to the line number of the "track changes" document.

**Comment 1: European Drought Directive and regional disparities**

You state "we advocate for a European Drought Directive..." but your results show significant regional disparities... It might be worth also caveating more this somehow?

We fully agree with this observation. While we support the idea of a harmonized European framework, we recognize the importance of allowing for regional variation in drought impact and management capacity. We have therefore clarified this in both the Introduction (Line 201-203 and Line 257) and Recommendations (Line 939) by explicitly stating that a European Drought Directive should balance harmonization with flexibility for regional adaptation needs.

**Comment 2: Abstract - mention of observational data**

In your data & results you also include observational data analysis... it might be worth mentioning this quantitative/meteorological data analysis in the abstract...

Thank you for pointing this out. We have revised the abstract to include the use of meteorological data, explicitly noting our application of the Standardized Precipitation-Evapotranspiration Index (SPEI) to assess drought conditions.

**Comment 3: Reference inconsistency in abstract**

The paper referred to in the abstract has a different title than the one mentioned in lines 88–89.

This was an oversight on our part. The reference has now been corrected throughout the manuscript.

**Comment 4: Clarify countries**

Perhaps just list a few example countries in the statement somewhere to help provide this context to the reader?

We have now included the countries that introduced restrictions during May–July 2022 according to the sources presented in that section. This change appears in Line 89.

**Comment 5: Inconsistent reference formatting**

Line 94: IPCCa – should this actually be IPCC, 2022a...?

We appreciate this catch. We corrected the formatting of these references in both the text and the reference list.

**Comment 6: Remove extra bracket**

Corrected as suggested.

**Comment 7: EC Flood Directive year inconsistency**

Was the Flood Directive established in 2007 or 2008?

We have verified and corrected the year to reflect consistent and accurate information (2007).

**Comment 8: Clarify spatial extent and time range of SPEI data**

We have now included a brief description of the CRU dataset, including its spatial resolution and temporal range (1950-2022), in Section 2.1 (Lines 278-283). We also justified the use of the 1971-2000 reference period in Lines 284-288.

**Comment 9: Typos and formatting**

Lines 224, 289, 330, 341 (old file) – Typo corrections requested.

We corrected all identified typos in these lines.

**Comment 10: Figure references and figure details**

Please signpost to specific figures and clarify time scales/definitions.

We have clarified the accumulation period for SPEI in Figure 4, now clearly labelled as SPEI-6, and reversed the order of the months as requested.

**Comment 11: Reorganize Section 3.1.2**

Consider merging or repositioning Section 3.1.2

Following your suggestion, we have moved and merged the paragraphs into a new subsection titled "Observed and Perceived Change in Drought" prior to Line 445.

**Comment 12: Figures 5 and 6 – clarify legends and colour schemes**

We have updated the caption of Figure 5 to explain white areas (no impacts) and revised the colour scheme of Figure 6 to better represent impact severity (see caption and figure).

**Comment 13: Figure 7 and associated text**

Clarify panel definitions, address 2% vs. 27% typo, low response rates, etc.

We corrected the typo to 27% (Line 540), clarified the meaning of panels a and b in the caption to Figure 7. The differences between the classification of the responses was also clarified.

**Comment 14: Length of captions for Figures 8 and 9**

Captions were shortened and harmonized in style. Redundant details about the survey response format was removed (see Figures 8 and 9 captions).

**Comment 15: Catalonia case study – strengthen links to data**

Can more of the responses from this region be explored...?

Yes. We significantly revised Section 3.4.1, explicitly linking Catalonia's drought onset to meteorological indicators (SPEI-6) and added detailed insights from local responses while bringing the length and structure of this section in line with the other spotlight case. We also stated the criteria for case study selection in Links 659-663, emphasizing Catalonia's and Italy's relevance due to high number of impacts, sufficient data, and institutional diversity.

**Comment 16: Boxed quote in Discussion**

Integrate into text and rephrase question contextually.

We reformatted the quote as part of the narrative and added clarifying context using brackets (Section 4.1.3).

**Comment 17: Referencing in Section 5.2**

We ensured that referencing in this section is now consistent with journal style.

**Comment 18: Conclusion – summarize key drought features**

As suggested, we now begin the Conclusion with a short summary of the key characteristics of the 2022 drought, reinforcing its significance for European drought policy.

**Reviewer 2 – Christian Massari**

We thank Dr. Massari for the thoughtful and positive comments throughout the review. We are especially grateful for your recognition of the value of combining empirical evidence with drought governance frameworks, and for the detailed suggestions that helped us significantly improve the precision and clarity of our methodology and conceptual framing. Below, we provide detailed responses to the comments made.

**Comment 1: Writing consistency and length**

Some sections would benefit from improved clarity, cohesion, and editorial refinement...

Thank you. We undertook a full language and structural review to improve consistency. Several sections were shortened or restructured to enhance clarity, including Section 2 and Section 3.

**Comment 2: Define key terms early**

I recommend defining terms like drought risk and drought impact earlier...

We agree and have added clear definitions of drought risk and drought impact in the first paragraph of the Introduction.

**Comment 3: Summary table of EU directives**

As suggested, we created Table 1 (Line 236) that summarizes the Water Framework Directive, EC Communications (2007, 2012), and the Floods Directive — including legal status, scope, and relevance for drought governance.

**Comment 4: Rename Section 2.1 and improve clarity**

The section is now titled "Climate Data and Drought Assessment". We improved the drought definition, adding SPEI threshold values (Lines 265-268), and described the CRU dataset, including the use of PET (Lines 277-283). We also clarified that PET is from pre-calculated CRU outputs.

**Comment 5: Regional vs. national focus**

Why center analysis on countries, when drought is a transboundary issue?

This is an important point. While drought is indeed transboundary, drought governance (e.g., plans, declarations, restrictions) is largely enacted at the national scale — making country-level analysis essential. Hence, we believe that as the focus of the study is on drought risk management, the country level information is key. However, we recognize that our current description of the regional averages may cause some confusion. Therefore, we clarified how regional aggregation was performed in Lines –343-354).

**Comment 6–8: Editorial clarity, citation adjustment, sentence rewording**

We have improved clarity, and relocated the citation of Avanzi et al. (2024) to Line 399, where it fits the narrative better.

**Comment 9: Clarify case study selection**

We now clearly state why Spain (Catalonia) and Italy were selected (Lines 659-663), based on the many impacts and impact severity, data sufficiency, and institutional variation. We also highlight how these cases illustrate broader themes from the survey.

**Comment 10: Decentralization and governance systems**

We expanded our discussion of governance systems in Lines 192-194, suggesting that decentralization might play a role in preparedness and coordination.

**Comment 11: Simplify Figure 6**

As advised, we have simplified Figure 6 to show only panel (a), focusing on impact severity. Prioritization results have been moved to a new Supplement Table S2. We also clarified the basis of the regional averages (Lines 29-342).

**Final Statement:**

We thank both reviewers again for their valuable and constructive feedback. The manuscript has been significantly improved as a result. We hope that the revised version meets your expectations and look forward to your final decision.

With kind regards, **Riccardo Biella**On behalf of all co-authors